# COLLABEDIT: TOWARDS NON-DESTRUCTIVE COLLABORATIVE KNOWLEDGE EDITING

**Jiamu Zheng** [1, §]   **Jinghuai Zhang** [3]   **Tianyu Du** [1, †]   **Xuhong Zhang** [1]   **Jianwei Yin** [1]   **Tao Lin** [2]

Zhejiang University [1]   Westlake University [2]   University of California, Los Angeles [3]
zhengjaamie@gmail.com   jinghuai1998@g.ucla.edu   zjradty@zju.edu.cn
zhangxuhong@zju.edu.cn   zjuyjw@cs.zju.edu.cn   lintao@westlake.edu.cn

## ABSTRACT

Collaborative learning of large language models (LLMs) has emerged as a new paradigm for utilizing private data from different parties to guarantee efficiency and privacy. Meanwhile, *Knowledge Editing (KE)* for LLMs has also garnered increased attention due to its ability to manipulate the behaviors of LLMs explicitly, yet leaves the collaborative KE case—in which knowledge edits of multiple parties are aggregated in a privacy-preserving and continual manner—unexamined. To this end, this manuscript dives into the first investigation of collaborative KE, in which we start by carefully identifying the unique three challenges therein, including *knowledge overlap*, *knowledge conflict*, and *knowledge forgetting*. We then propose a non-destructive collaborative KE framework, COLLABEDIT, which employs a novel model merging mechanism to mimic the global KE behavior while preventing the severe performance drop. Extensive experiments on two canonical datasets demonstrate the superiority of COLLABEDIT compared to other destructive baselines, and results shed light on addressing three collaborative KE challenges and future applications. Our code is available at https://github.com/LINs-lab/CollabEdit.

## 1   INTRODUCTION

Large Language Models (LLMs) (Achiam et al., 2023; Qiao et al., 2023) recently have emerged as the promising solution toward general artificial intelligence. However, deploying LLMs in practice usually requires customizing LLMs with specific knowledge (Meng et al., 2022), where re-training LLMs may be expensive and unacceptable (Jang et al., 2023). Accordingly, Knowledge Editing (KE) (Meng et al., 2022; Mitchell et al., 2022a; Tan et al., 2024; Zhang et al., 2023), which allows efficient modification of knowledge stored in existing models, has been proposed as an alternative solution.

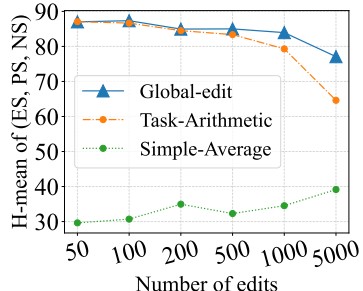

Figure 1: Limits of existing KE methods under the collaborative KE scenarios on the Multi-CounterFact dataset (Meng et al., 2022).

To explicitly update LLMs with knowledge from multiple parties or organizations—each possesses a distinct and private dataset (Ye et al., 2024; Wu et al., 2022; McMahan et al., 2017)—and meet individual demands, current KE methods (Meng et al., 2022; 2023) first need to collect edit requests from these parties with violated privacy concerns: the edit request itself contains sensitive private information and thus becomes infeasible for sharing. It motivates resorting to the cross-silo collaborative learning paradigm (Wu et al., 2023; Kairouz et al., 2021)—by only communicating the locally-updated-models, rather than uploading a list of risky edit requests—namely collaborative KE for LLMs.

However, existing KE methods are all designed for the single-party single-model scenario (Meng et al., 2022; Mitchell et al., 2022a; Tan et al., 2024; Meng et al., 2023). Noting that model merging (MM) techniques (Ortiz-Jimenez et al., 2023; Chronopoulou et al., 2023; Yadav et al., 2023) allow a straightforward extension of KE methods to a collaborative KE scenario. Therefore as our

---

§Work was done during Jiamu's visit to Westlake University.

†Corresponding author.

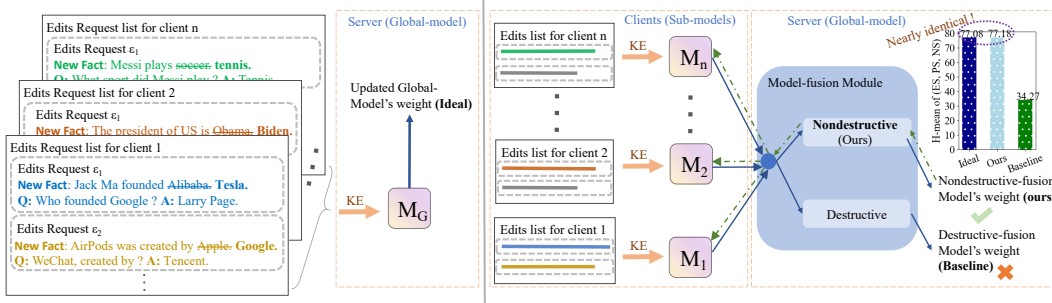

(a) Global Editing            (b) Collaborative Editing

Figure 2: Comparison of global KE (GLOBAL-EDIT) and collaborative KE.

(side)-contribution, we examine a naive combination of local KE and MM techniques, and compare them with the optimal global KE method (GLOBAL-EDIT): we can witness that all these naive collaborative KE methods are destructive. In detail, we conduct independent KE (i.e., MEMIT (Meng et al., 2023)) on each party locally, and then use model merging techniques like Simple-Average (Chronopoulou et al., 2023) and Task-Arithmetic (Ilharco et al., 2023) to merge local models (LLMs) into a global model (LLM). We find that as the number of edits increases, the performance gap between naive baselines with the optimal GLOBAL-EDIT also widens.

Alongside the pitfalls of naive collaborative KE methods, in this manuscript, we carefully examine the intervention issues among different parties and identify the *knowledge overlap* and *knowledge conflict* challenges. These challenges arise from the global server's blind access to the edit requests of each party. In addition, the continual KE requirement for each party inherently results in the interventions among different rounds of editing, corresponding to *knowledge forgetting* challenge.

To bridge the gap and provide a deeper understanding of collaborative KE, we first analyze the performance drop between naive collaborative KE methods and GLOBAL-EDIT from a theoretical perspective, upon which we design a novel framework COLLABEDIT that allows non-destructive collaborative KE. We further pioneer the explorations on the interventions associated with collaborative KE (namely the three challenges we identified) and design tailored solutions to effectively address them. **Our contributions can be summarized as follows:**

- We are the first to propose the collaborative KE paradigm (including naive collaborative KE baselines, GLOBAL-EDIT, and our COLLABEDIT), in which we summarize the unique interventions associated with collaborative KE and conclude them with three challenges in this novel paradigm.
- We identify the performance gap between the naive collaborative KE method and the upper bound performance (i.e., GLOBAL-EDIT) through theoretical and empirical analysis.
- To the best of our knowledge, we propose the first non-destructive collaborative KE framework: it is versatile, allowing nondestructive integration of existing KE methods and providing insights into the solution of each challenge.
- Our empirical results demonstrate the effectiveness of our proposed framework compared with baselines and that of the novel solutions to three challenges based on our COLLABEDIT. Our discussions shed light on future research for collaborative KE.

## 2 RELATED WORK

**Knowledge Editing (KE).** KE (De Cao et al., 2021; Mitchell et al., 2022a) has received significant attention due to the increasing demands for efficient updating of LLMs. Hypernetwork knowledge editing and direct model editing are the two most representative KE methods. Given edit requests, hypernetwork knowledge editing (Mitchell et al., 2022a;b; Tan et al., 2024) leverages a trained hypernetwork to predict the model updates, while direct model editing (Meng et al., 2023; 2022; De Cao et al., 2021) updates LLMs as an associative memory and inserts new memory via solving an optimization problem. However, these KE methods are all designed for a single LLM, which limits their applications to a more practical collaborative learning scenario. In this paper, we place emphasis on the SoTA frameworks of two KE methods types stated above, which are respectively MALMEN (Tan et al., 2024) and MEMIT (Meng et al., 2023), to explore the integration of KE and collaborative learning.

**Collaborative learning and model merging.** Collaborative learning (Kairouz et al., 2021; Wang et al., 2021; Fan et al., 2024; Mohtashami et al., 2023) allows multiple parties to jointly and continuously learn a machine learning model by sharing their updates to a global server for aggregation. Alongside the orthogonal techniques to address data heterogeneity issue (Karimireddy et al., 2020; Li et al., 2019), model aggregation/merging (Li et al., 2023b; Wortsman et al., 2022; Ortiz-Jimenez et al., 2023; Yadav et al., 2023) has emerged as a promising research direction to collaborative learning, which employs the global server to directly merge model updates in the weight space without disclosing the training data of each party. The most commonly used model merging techniques themselves are Simple-Average (SA) (Chronopoulou et al., 2023; Wortsman et al., 2022) and Task-Arithmetic (TA) (Ortiz-Jimenez et al., 2023). Moreover, TIES-merging (Yadav et al., 2023) has recently proposed to further enhance the merging performance by solving the symbol conflicts among different models. However, all existing model merging techniques only achieve destructive editing performance when used for collaborative KE, which inevitably results in knowledge loss during the merging process.

## 3 PRELIMINARIES OF COLLABORATIVE KNOWLEDGE EDITING

We first introduce the basics of KE in a single LLM. Then, we illustrate the naive approaches to conduct collaborative KE. Finally, we describe the inherent interventions within collaborative KE.

### 3.1 INTRODUCTION TO KNOWLEDGE EDITING IN A SINGLE LLM

LLMs can answer natural-language queries about *facts* based on implicit knowledge encoded within the parameters. Following Meng et al. (Meng et al., 2023), we define a fact $f$ as "*(subject s, relation r, object o)*", e.g., "*(s = Danielle Darrieux, r = spoke the language, o = French)*". Given a sequence of facts $\mathcal{E} = \{f_i | f_i = (s_i, r_i, o_i)\}$ to edit (denoted as *edit requests*), knowledge editing aims to maximize the likelihood that the updated LLM $\mathcal{M}_{\boldsymbol{\theta}}$ predicts the desired object $o_i$ for any factual prompt $\mathbf{x} \oplus p(s_i, r_i)$, which involves a prefix $\mathbf{x}$ and a templated prompt $p(s_i, r_i)$:

$$\arg\min_{\mathcal{M}_{\boldsymbol{\theta}}} \frac{1}{|\mathcal{E}|} \sum_{i=1}^{|\mathcal{E}|} \mathbb{E}_{\mathbf{x}} \left[ -\log \Pr_{\mathcal{M}_{\boldsymbol{\theta}}} \left[ o_i | \mathbf{x} \oplus p(s_i, r_i) \right] \right]. \tag{1}$$

The state-of-the-art knowledge editing methods (Meng et al., 2022; 2023; Tan et al., 2024) found that modifying a small sequence of MLP layers in the critical path of LLM is sufficient to edit its factual associations. In particular, linear operation $\mathbf{W}^l$ in an MLP layer can operate as a key-value store for input keys $\mathbf{K}^l$ and the memory/knowledge values $\mathbf{M}^l$, where input keys correspond to the intermediate feature vector of the model from a set of edit requests. Knowledge editing modifies each MLP layer such that it associates $\mathbf{K}^l$ to the desired $\mathbf{M}^l$ by solving $\mathbf{W}^l \mathbf{K}^l \approx \mathbf{M}^l$. For brevity, we will describe knowledge editing for a specific layer and omit $l$ throughout the paper.

Given a set of facts $\mathcal{E}$ to edit (i.e., edit requests), we first obtain their input keys $\mathbf{K} = [\mathbf{k}_1, \ldots, \mathbf{k}_{|\mathcal{E}|}]$ to the layer $l$ via a single feed-forward. We also obtain the desired memory values $\mathbf{M} = [\mathbf{m}_1, \ldots, \mathbf{m}_{|\mathcal{E}|}]$ of layer $l$ that maximize $\Pr[o_i | \mathbf{x} \oplus p(s_i, r_i)]$. The goal of editing the layer $l$ can be formulated as optimizing the $\boldsymbol{\Delta}$ such that the updated weight $\mathbf{W} + \boldsymbol{\Delta}$ associates the input keys $\mathbf{K}$ to the desired memory values $\mathbf{M}$. Note that the MLP layer also contains previously stored memories of existing knowledge, which should be preserved during the knowledge editing. Therefore, we also maintain the associations between input keys of existing knowledge $\mathbf{K}_{\text{init}}$ and their memory values ($\mathbf{W}\mathbf{K}_{\text{init}}$). Following MEMIT (Meng et al., 2023), we derive the closed form of $\boldsymbol{\Delta}$ for a specific layer $l$ as:

$$\boldsymbol{\Delta} = \mathbf{R}\mathbf{K}^\top (\mathbf{C} + \mathbf{K}\mathbf{K}^\top)^{-1}, \tag{2}$$

where $\mathbf{C} = \mathbf{K}_{\text{init}}\mathbf{K}_{\text{init}}^\top$ is the covariance matrix of the input keys of existing knowledge, and $\mathbf{R} = \mathbf{M} - \mathbf{W}\mathbf{K}$ represents the residual error in the output space of layer $l$. See more details in Appendix A.

### 3.2 DESTRUCTIVE MODEL MERGING ENCOUNTERS KNOWLEDGE EDITING

KE in practice involves editing the factual associations of LLM, such as correcting the hallucinations or updating outdated information. This process often requires handling simultaneous edit requests, where multiple parties or clients access and collaboratively contribute to the same LLM service. Though *Global KE* (GLOBAL-EDIT) illustrated in Figure 2(a) represents the ideal editing cases, it also necessitates each client to directly share the edit requests with the server, which violates the privacy constraints. *Collaborative KE* in Figure 2(b) instead allows each client to edit on its

local model and only rely on the server to aggregate the edit updates using the model merging techniques (Wortsman et al., 2022; Ortiz-Jimenez et al., 2023; Yadav et al., 2023).

However, existing KE algorithms are all designed for a single client and cannot be trivially generalized to the collaborative KE scenario. As evidenced in Figure 1, naively extending existing editing methods or model merging methods yields a dramatic performance drop compared to that of the GLOBAL-EDIT (upper bound), especially when the number of edits increases. Given the limits of *destructive collaborative KE* methods, we aim to develop a *non-destructive collaborative KE* method that can achieve a similar editing performance as GLOBAL-EDIT, even with a large number of edits.

### 3.3 INTERVENTIONS WITHIN COLLABORATIVE KNOWLEDGE EDITING

In addition to the performance drop, our proposed concept of collaborative KE also suffers from several key challenges, due to the unique characteristics of this scenario. By default, we assume a trustworthy and non-adversarial collaborative KE scenario. The collaborative KE employs a global server to aggregate the edits of local clients without disclosing their edit requests, while requiring each client to continually edit the global model by updating its local model in a multi-round manner. However, there still exist several unique challenges due to the interventions among different clients and different rounds of editing, warranting research in the future.

#### 3.3.1 INTERVENTIONS AMONG DIFFERENT CLIENTS

In collaborative KE, multiple clients may use similar edit requests to update their local models and send the updated models to the global server for aggregation. The interventions among clients are then raised in the editing event $e := (s_1, r_1, o_1 \rightarrow o_2, t_1, m_1)$, which includes $s_1$ as a **s**ubject, $r_1$ as a **r**elationship, $o_1$ and $o_2$ as **o**bjects, $t_1$ as a editing **t**imestamp and $m_1$ as client **m**odel.

**Knowledge conflict** indicates that edit requests from the same/different clients (in the same round[1]) share the same subject $s$ and relation $r$ but with different objects $o$. Such a conflict renders the effectiveness of knowledge editing and may even compromise the overall KE performance. We elaborate the general formulation of *conflict edit* below (detailed illustration can be found in the Table 7 of Appendix):

$$\begin{cases} e_1 = (s_1, r_1, o_1 \rightarrow o_2, t_1, m_1) \\ e_2 = (s_1, r_1, o_1 \rightarrow o_3, t_2, m_2) \end{cases}, \tag{3}$$

where local model $m_1$ and $m_2$ perform a conflicting editing for the same subject $s_1$ and relationship $r_1$ at timestamp $t_1$ and $t_2$ respectively, changing the same original object $o_1$ to different $o_2$ and $o_3$.

Similar to the composite edit operations mentioned by Li et al. (2024), composite conflict and composite overlap arising from such operations may also occur in collaborative KE scenarios, with even more diverse and complex forms. Here we aim to briefly introduce the key concept of knowledge conflict, and a more detailed definition and investigation of this issue left for future work.

**Knowledge overlap** is a simplified case of knowledge conflict, where the object changing relationship (i.e. $o_1 \rightarrow o_2$ and $o_1 \rightarrow o_3$) in editing events of $e_1$ and $e_2$ becomes identical. Knowledge overlap is also closely related to the overfitting problem in machine learning, in which excessive overlapped editing requests can degrade the model's editing performance on other edit requests (excluding those repeated edit requests).

#### 3.3.2 INTERVENTIONS AMONG DIFFERENT ROUNDS OF EDITING

The collaborative KE paradigm naturally requires multiple clients to continually update their local models in a multi-round manner and thus edit the global model with the latest knowledge. **Knowledge forgetting** issue, therefore, arises given the continual arrival of a large number of new editing requests, alongside the existing knowledge and editing requests.

Assume that each client has a set of old edit requests $\mathcal{E}_o$, as well as $m$ sets of new edit requests $\mathcal{E}_n = [\mathcal{E}_{n_1}, \mathcal{E}_{n_2}, \cdots, \mathcal{E}_{n_m}]$, where the new edit requests are irrelevant (i.e., their subjects $s$ and relationships $r$ are different) to the old edit requests. The model is initialized by updating the model with the old edit requests $\mathcal{E}_o$, and the local model of each client will be updated with the new edit requests $\mathcal{E}_{n_i}$ at $i$-th round of editing, followed by the model aggregation step. The knowledge

---

[1]In cases of conflict between edits from different rounds, due to the overwriting nature of KE, the latter conflicting edit will overwrite the former, naturally resolving the conflict.

forgetting issue encountered after $m$ rounds of local editing and global aggregation can then be defined as *the editing performance on the old knowledge obtained from the old edit requests $\mathcal{E}_0$.*

In particular, we find that as the value of $m$ increases, the evaluation performance of the model on old knowledge $\mathcal{E}_o$ deteriorates, as evidenced in Section 5.3.

## 4 METHODOLOGY

### 4.1 COLLABEDIT: NON-DESTRUCTIVE COLLABORATIVE KE

To better understand the performance drop, we first explicitly model the relationship between the weight updates $\boldsymbol{\Delta}_G$ of the global model using GLOBAL-EDIT and that of each client model $\boldsymbol{\Delta}_i$ using local editing. For ease of presentation, we consider the collaborative KE scenario with $N$ clients and each client model has $M$ edit requests. We simplify the theoretical analysis to the single-round editing case and demonstrate the effectiveness of COLLABEDIT for multi-round editing in Remark 2.

**Lemma 1** (The relationship between the weight updates from GLOBAL-EDIT and local editing). *Take the KE method MEMIT as an example. Following the definitions in Section 3.1, we denote $\mathbf{C}$ as an aggregated statistic over the previously stored keys of existing knowledge and use $\mathbf{K}_i$ to represent the new keys derived from client $i$'s edit. Then, the relationship between $\boldsymbol{\Delta}_G$ and $\boldsymbol{\Delta}_i$ is measured as:*

$$\boldsymbol{\Delta}_G = \sum_{i=1}^N \boldsymbol{\Delta}_i \cdot \left( \alpha_i := (\mathbf{C} + \mathbf{K}_i\mathbf{K}_i^\top)(\mathbf{C} + \sum_{i=1}^N \mathbf{K}_i\mathbf{K}_i^\top)^{-1} \right). \tag{4}$$

*See detailed proof in Appendix B.1.*

**Intuition:** If we can estimate $\boldsymbol{\Delta}_G$ using $\boldsymbol{\Delta}_i$, then we can merge $\{\boldsymbol{\Delta}_i\}_{i=1}^N$ to obtain the same global model as GLOBAL-EDIT and, therefore, obtain non-destructive collaborative KE.

**Details of COLLABEDIT:** Indeed $\boldsymbol{\Delta}_G$ can be represented as the weighted sum of different local weight updates $\boldsymbol{\Delta}_i$ with coefficient $\alpha_i$. However, the coefficient $\alpha_i$ relies on the value of $\mathbf{K}_i$ of all the clients: it breaks the privacy, given the fact that $\mathbf{K}_i$ is an intermediate feature vector of the model from a set of edit requests and any external party can easily reconstruct the edit requests if $\mathbf{K}_i$ is leaked. As a remedy, our COLLABEDIT instead proposes to directly communicate $\mathbf{K}_i\mathbf{K}_i^\top$, in which we prove in Section 6 that $\mathbf{K}_i\mathbf{K}_i^\top$ is non-trivial to attack. See our pseudo-code in Appendix D.

**Remark 1.** *Currently, we consider two mainstream KE methods (Akyürek et al., 2023), namely (1) locate and edit activations (same as "Direct model editing" mentioned in Section 2, e.g., MEMIT (Meng et al., 2023) and ROME (Meng et al., 2022)); and (2) train an auxiliary model to directly predict parameters (same as "Hypernetwork knowledge editing" mentioned in Section 2, e.g., MEND (Mitchell et al., 2022a) and MALMEN (Tan et al., 2024)). Our framework COLLABEDIT is general enough to integrate many other KE methods, and we leave them for future work.*

**Justifying the performance drop for destructive editing approaches.** We further analyze the performance degradation for destructive editing approaches when the number of edits increases, as illustrated in Figure 1. For the sake of simplicity, we take the TASK-ARITHMETIC (Ilharco et al., 2023) with MEMIT as an example. The drop can be explained by:

$$\boldsymbol{\Delta}_G - \boldsymbol{\Delta}_G' = \sum_{i=1}^N \boldsymbol{\Delta}_i \left[ (\mathbf{C} + \mathbf{K}_i\mathbf{K}_i^\top)(\mathbf{C} + \sum_{j=1}^N \mathbf{K}_i\mathbf{K}_i^\top)^{-1} - \lambda\mathbf{I} \right], \tag{5}$$

where $\boldsymbol{\Delta}_G$ and $\boldsymbol{\Delta}_G'$ represent the weight updates derived from COLLABEDIT (our non-destructive collaborative KE) and a destructive collaborative KE using TASK-ARITHMETIC, respectively. We can see that the impact of new knowledge $\mathbf{K}_i\mathbf{K}_i^\top$ is negligible compared to existing knowledge $\mathbf{C}$ when the number of edits is small[2], resulting in $(\mathbf{C} + \sum_{j=1}^N \mathbf{K}_i\mathbf{K}_i^\top)^{-1} \approx \mathbf{C}$ and thus $\boldsymbol{\Delta}_G \approx \boldsymbol{\Delta}_G'$ when $\lambda = 1$. The gap becomes wider when the number of edits increases, contributing to the continuous decline in TASK-ARITHMETIC's performance in Figure 1 compared to GLOBAL-EDIT.

**Remark 2** (COLLABEDIT is effective for multi-round editing). *Collaborative KE involves multiple clients continuously editing the local models and sharing the updated global model across multiple rounds, and thus requires robust support to ensure seamless knowledge integration and consistent knowledge memorization. COLLABEDIT achieves non-destructive collaborative KE for single-round editing—as an approximation of aggregating all edit requests of clients in a specific round and applying global KE to update the global model—remains effective for multi-round editing. Note that multi-round editing is equivalent to applying global KE to iteratively update a single LLM multiple times under the reasonable editing budgets (Gupta et al., 2024).*

---

[2]We randomly sample 100 edit requests to estimate the norm of $K_iK_i^T$. We observe that the average $\ell_2$-norm of $K_iK_i^T$ is approximately $0.0001\%$ of that of $C$, which supports the claim.

### 4.2 REMEDY TOWARDS SOLVING INTERVENTION CHALLENGES IN COLLABORATIVE KNOWLEDGE EDITING: SOME CASE STUDIES

Interventions within collaborative KE scenarios are non-trivial, due to the challenges of explicitly modeling the impacts of editing requests from different clients. Our COLLABEDIT paves the path by mimicking the optimal GLOBAL-EDIT and allowing the non-destructive editing. This subsection case studies how our COLLABEDIT sheds insights on solving unique challenges caused by the interventions among different clients (spatial aspect) and editing rounds (temporal aspect), namely knowledge overlap, knowledge conflict, and knowledge forgetting.

#### 4.2.1 EDITING RESIDUAL DETECTS KNOWLEDGE OVERLAP

COLLABEDIT simplifies the knowledge overlap challenge in collaborative KE scenarios into the over-fitting problem under the global KE scenarios. In other words, multiple clients edit the same piece of knowledge is equivalent to integrating several identical pieces of knowledge into the global model. In detail: performing KE in the model results in weights update $\mathbf{\Delta}$ and residual $\mathbf{R}_{\text{old}}$, as determined by the input key $\mathbf{K}$. In the case of editing the same knowledge (i.e., same $\mathbf{K}$), we can get new residual $\mathbf{R}_{\text{new}} = \mathbf{R}_{\text{old}} - \mathbf{\Delta K}$, where the following equation can be leveraged to track the dynamics of KE:

$$\mathbf{R}_{\text{new}} := \mathbf{R}_{\text{old}} - \mathbf{\Delta K} = \mathbf{R}_{\text{old}} - \mathbf{R}_{\text{old}}\mathbf{K}^\top(\mathbf{C} + \mathbf{K}\mathbf{K}^\top)^{-1}\mathbf{K}. \tag{6}$$

Intuitively, (6) explains that the residual should gradually approach $0 \cdot \mathbf{I}$. If the residual $\mathbf{R}$ gradually approaches zero, then we can accurately detect the knowledge overlap by examining the residual $\mathbf{R}$, as demonstrated in Section 5.3.

#### 4.2.2 ADDRESSING KNOWLEDGE CONFLICT VIA DATA AUGMENTAION

Recall that in rare cases, edit requests from the same/different clients in the same round may share the same subject $s$ and relation $r$ but with different objects $o$, known as knowledge conflict. An ideal solution to the knowledge conflict should consist of two stages. In the first stage, the global server and clients need to collaboratively detect the conflict in a privacy-preserving manner. For example, when the knowledge conflict occurs, the global server produces poor editing performance on some edit requests. As a result, the clients (who contribute to the edits) could report the issue.

Once the conflict is identified, the server will determine which of the conflicting edit requests to retain for the global model based on the client's report and a predefined strategy (e.g., FCFS (Zhao & Stankovic, 1989) or FIFO (Morse & Richardson, 1983) strategy). The client whose edit request is selected for integration can apply data augmentation techniques, such as incorporating relevant knowledge (Li et al., 2024), to enhance the KE of the selected edit request and effectively resolve the knowledge conflict.

#### 4.2.3 DYNAMIC COVARIANCE MATRIX ALLEVIATES KNOWLEDGE FORGETTING

The previously memorized knowledge may be forgotten by the LLM after a large number of edits, termed as *knowledge forgetting* issue. COLLABEDIT simplifies the analysis of this issue and we can witness from (2) that the covariance matrix $\mathbf{C}$ of existing knowledge is immutable, amplifying the forgetting as the number of edits increases. As a remedy, we propose using a dynamic version of $\mathbf{C}$, i.e.,

$$\mathbf{C} = \beta_0\mathbf{C}_0 + \beta_1\mathbf{C}_1 = \beta_0\mathbf{C}_0 + \beta_1\sum\mathbf{K}_i\mathbf{K}_i^\top, \tag{7}$$

where $\beta_0$ and $\beta_1$ are hyper-parameters that balance the influences of existing knowledge and newly acquired knowledge. $\mathbf{C}_0$ is the covariance matrix of existing knowledge and $\mathbf{C}_1$ is the accumulated covariance matrix of new knowledge. $\mathbf{K}_i$ represents the input keys obtained from all the edit requests at the $i$-th round. The dynamic covariance matrix continuously updated for the new knowledge can effectively mitigate the knowledge forgetting issue, as verified in Section 5.3.

## 5 EXPERIMENTS

### 5.1 EXPERIMENTAL SETUP

**Datasets and models.** Following the literature (Meng et al., 2022; 2023), we use Multi-CounterFact (MCF) (Meng et al., 2022) and zsRE (Levy et al., 2017) as datasets and evaluate the editing performance on GPT2-XL (Radford et al., 2019) and GPT-J (6B) (Wang & Komatsuzaki, 2021).

Table 1: Overall editing performance on GPT2-XL. GLOBAL-EDIT is 5000 × 1, which means we edit 5000 requests in one model (global model) at one time. GLOBAL-EDIT is an ideal situation. Others are merging methods (500 × 10) where we edit 10 models and each model will be edited by 500 requests. The line of GPT2-XL means we directly evaluate 5000 requests without any editing operation to test the model's original performance. The "Score" serves as the overall metric for assessing the performance of each method on each dataset.

| Method | MCF | | | | zsRE | | | |
|---|---|---|---|---|---|---|---|---|
| | NS ↑ | PS ↑ | ES ↑ | Score ↑ | NA ↑ | PA ↑ | EA ↑ | Score ↑ |
| GPT2-XL | 78.24 | 23.88 | 21.50 | 29.65 | 24.32 | 21.87 | 22.80 | 22.95 |
| GLOBAL-EDIT | 65.08 | 80.66 | 89.66 | 77.08 | 25.25 | 64.71 | 68.96 | 43.12 |
| TIES-MERGING | **78.46** | 26.35 | 27.16 | 34.27 | 24.94 | 25.99 | 27.59 | 26.12 |
| TASK-ARITHMETIC | 66.84 | 55.19 | 61.66 | 60.85 | 24.97 | 33.66 | 34.80 | 30.45 |
| SIMPLE-AVERAGE | 76.90 | 29.97 | 33.06 | 39.15 | **25.78** | 29.26 | 30.62 | 28.40 |
| COLLABEDIT | 65.26 | **80.67** | **89.70** | **77.18** | 25.21 | **64.27** | **68.40** | **42.95** |

Table 2: Overall editing performance on GPT-J (6B), based on MEMIT (Meng et al., 2023). The experimental setting is identical to GPT2-XL in Table 1. The "Score" serves as the overall metric.

| Method | MCF | | | | zsRE | | | |
|---|---|---|---|---|---|---|---|---|
| | NS ↑ | PS ↑ | ES ↑ | Score ↑ | NA ↑ | PA ↑ | EA ↑ | Score ↑ |
| GPT-J | 83.45 | 17.17 | 14.78 | 21.75 | 26.99 | 26.25 | 27.04 | 26.75 |
| GLOBAL-EDIT | 57.20 | 96.13 | 99.26 | 79.03 | 28.05 | 88.79 | 92.05 | 51.92 |
| TIES-MERGING | 76.15 | 30.13 | 30.98 | 38.16 | **30.17** | 42.55 | 43.55 | 37.68 |
| TASK-ARITHMETIC | 50.24 | 72.82 | 73.26 | 63.44 | 18.77 | 45.16 | 46.75 | 30.98 |
| SIMPLE-AVERAGE | **78.04** | 41.28 | 54.68 | 54.22 | 29.19 | 47.96 | 51.38 | 40.22 |
| COLLABEDIT | 57.12 | **96.03** | **99.06** | **78.91** | 28.26 | **88.78** | **92.19** | **52.17** |

**Baselines.** We compare COLLABEDIT with three naive collaborative KE methods, which apply standard KE algorithms (e.g., MEMIT (Meng et al., 2023) and MALMEN (Tan et al., 2024)) to update the local model and use the current model merging algorithms to merge local updates into the global model. In particular, we experiment with three most commonly used algorithms for model merging, including SIMPLE-AVERAGE (Chronopoulou et al., 2023), TASK-ARITHMETIC (Ortiz-Jimenez et al., 2023), and TIES-MERGING (Yadav et al., 2023).

**Evaluation metrics.** Unless otherwise mentioned, we utilize MEMIT as the backend KE algorithm and adopt the same metrics as MEMIT to evaluate editing performance. Strictly following the literature (Meng et al., 2022; 2023; Tan et al., 2024), we use Efficacy Score (ES), Paraphrase Score (PS), Neighborhood Score (NS), N-gram Entropy (NE), Reference Score (RS), and Score (i.e., the harmonic mean of ES, PS, NS) as metrics for MCF; we use Neighborhood Accuracy (NA), Paraphrase Accuracy (PA), Efficacy accuracy (EA), and Score (i.e., the harmonic mean of NA, PA, and EA) as metrics for zsRE. When using MALMEN (Tan et al., 2024) as the backend KE algorithm, we adopt the same metrics as MALMEN for a fair comparison, including "editing success" (EA), "generalization success" (PA), and "locality success" (NA). See detailed descriptions in Appendix C.

**Evaluation benchmark for conflict knowledge editing scenarios.** In order to conveniently simulate potential scenarios of collaborative knowledge conflict and analyze the issues and impacts that these scenarios may bring, we reconstruct two existing benchmarks to simulate knowledge conflict situations through GPT-3.5-turbo. Initially, we attempt to explore the impact of knowledge conflict on model performance using Multi-CounterFact (MCF) (Meng et al., 2022) due to its large scale. For each data point $(s, r, o)$ in the MCF dataset, we utilize GPT-3.5-turbo to generate a conflict object that is identical to $s$ and $r$ but differs in $o$. Section E in the Appendix showcases a concrete example of the generated conflict object. To validate the effectiveness of our two-stage mechanism to resolve knowledge conflict, we utilize the Easy dataset (Li et al., 2024) for the sake of simplicity. This dataset was constructed by creating several additional related knowledge edits for each edit using Wikipedia as the source, which MCF does not include. Additionally, we also generated a corresponding conflict object for each edit in the dataset using GPT-3.5-turbo.

Table 3: Overall editing performance on GPT-J (6B) and GPT2-XL, based on MALMEN (Tan et al., 2024). We edit 8 models and each model will be edited by 125 requests of zsRE. The "Score" serves as the overall metric.

| Method | GPT2-XL (zsRE) | | | | GPT-J (zsRE) | | | |
|---|---|---|---|---|---|---|---|---|
| | EA ↑ | PA ↑ | NA ↑ | Score ↑ | EA ↑ | PA ↑ | NA ↑ | Score ↑ |
| GLOBAL-EDIT | 99.21 | 93.08 | 16.5 | 36.84 | 99.95 | 95.66 | 27.32 | 52.57 |
| TIES-MERGING | 15.52 | 14.85 | 18.68 | 16.18 | 27.86 | 26.76 | 25.18 | 26.55 |
| TASK-ARITHMETIC | 50.37 | 45.79 | 4.28 | 10.89 | 27.59 | 29.6 | 26.3 | 27.76 |
| SIMPLE-AVERAGE | 52.39 | 46.03 | 4.58 | 11.57 | 71.15 | 53.96 | 4.82 | 12.49 |
| COLLABEDIT | 99.06 | 92.66 | 15.49 | 35.11 | 99.62 | 92.88 | 23.25 | 47.01 |

## 5.2 EXPERIMENTAL RESULTS OF KE PERFORMANCE

**Superior collaborative knowledge editing performance.** As shown in Table 1 and 2 when using MEMIT (Tan et al., 2024) as the backend KE algorithm, our privacy-preserving solution COLLABE-DIT achieves on-par editing performance with that of GLOBAL-EDIT, and significantly outperforms other naive model merging methods in terms of the "Score" on two datasets and two models. Additionally, our COLLABEDIT has nearly identical performance with GLOBAL-EDIT via combining the weight updates of each client, which ensures both privacy protection and editing quality. Nevertheless, there exists a significant gap between the performance of baselines and GLOBAL-EDIT. Table 3 additionally shows the editing performance of COLLABEDIT when using MALMEN as the backend KE algorithm (Tan et al., 2024): COLLABEDIT is capable of performing nondestructive collaborative KE across various mainstream KE methods.

**Discussion about the performance of baselines.** Though other baselines (Table 1 and 2) have a relatively higher NS value compared to GLOBAL-EDIT and our COLLABEDIT, we conjecture that it might be caused by the under-fitting phenomenon: these model merging methods are not specifically designed for merging the weight updates from knowledge editing, which is reflected by their low values of PS, ES, and Score. The results of two models GPT2-XL and GPT-J (6B) (the first line) further confirms that the high NS of other baselines are largely due to the inherent high quality of the model itself, exhibiting their poor collaborative KE effects. Note that NS/NA emphasizes that the edited model should maintain the same answer for neighborhood prompts of edit requests. However, editing certain knowledge using existing KE (Tan et al., 2024; Meng et al., 2023) methods would inevitably affect the association of its neighboring prompts, which leads to a similar drop of NS/NA for both GLOBAL-EDIT and COLLABEDIT.

## 5.3 EXPERIMENTAL RESULTS ON THREE CHALLENGES OF COLLABORATIVE KE

**Residual R can effectively detect the knowledge overlap.** To understand the impacts of knowledge overlap, we repeatedly edit the same edit requests into the global model. Figure 3 shows that as the number of repeating edits increases, the $\ell_2$-norm of residual $\mathbf{R}$ reduces rapidly and becomes smaller than $0.01$ when repeating edits for 12 times, which is consistent with our theoretical analysis in Section 4.2.1. This implies that the $\ell_2$-norm of $\mathbf{R}$ can be used to check whether "overlapped editing" happens, which may be helpful for practitioners to avoid the decrease in model performance.

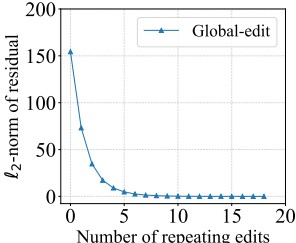

Figure 3: The $\ell_2$-norm of residual $\mathbf{R}$ when data replication happens.

**Knowledge conflict can compromise the editing performance.** To explore the impact of knowledge conflict, we reconstruct MCF with knowledge conflict (see Section 5.1 for details), where each edit request $f' = (s', r', o')$ in the benchmark corresponds to a $f = (s, r, o)$ in MCF and $s' = s$, $r' = r$, $o' \neq o$ (based on the definition in Section 4.2.2). We randomly sample 5,000 edit requests and their conflicted versions from both datasets, denoted as $\mathcal{E}$ and $\mathcal{E}'$. For experiments, we can either distribute edit requests in $\mathcal{E}$ or both sets ($\mathcal{E}$ and $\mathcal{E}'$) across all the clients for collaborative KE to understand the impact of $\mathcal{E}'$ on $\mathcal{E}$. Table 6 (see Appendix) evaluates the KE performance of $\mathcal{E}$ with and without the editing of the conflicted set $\mathcal{E}'$ to explore the impact of knowledge conflict on KE performance. We can see that *the overall KE performance largely decreases due to conflicting knowledge especially for PA and EA*: as those accuracy-related metrics, in comparison to the success-related

metrics (i.e., PS, NS, ES), are more rigorous; while NA, a metric used to assess whether irrelevant knowledge is affected, nearly remained unchanged. See Section C in Appendix for details.

**Two-stage mechanism with knowledge augmentation can mitigate conflicts.** Given the harmful impacts of knowledge conflict, we examine our two-stage mechanism (introduced in section 4.2.2) on the modified Easy dataset (Li et al., 2024). In this scenario, there is no objective standard to determine which edit should be retained. Therefore, we can employ the FCFS (Zhao & Stankovic, 1989) or FIFO (Morse & Richardson, 1983) to select the correct edit to be preserved. Subsequently, we augment edit requests and obtain weight updates from the selected client.

Firstly, we present a detailed example of resolving knowledge conflicts in Figure 4. Specifically, given the question *"What use does 'fpart' have?"*, there are two edit requests that induce conflicting answers, i.e., "data migration" and "data transfer". Let's define "data migration" as the target knowledge to preserve and "data transfer" as the conflict knowledge to remove, and we have the following observations after adopting the proposed mechanism: (1) Before solving the conflict (left), the LLM produces a large output probability for both "data migration" and "data transfer"; (2) After solving the conflict (right), the probability of "data migration" slightly increases while the probability of "data transfer" drops to 0. Moreover, the probability of unrelated knowledge remains unchanged. The results show that knowledge conflict is effectively resolved.

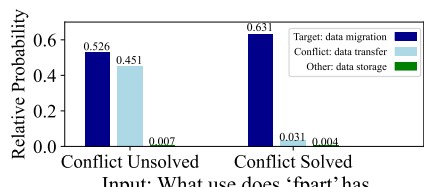

Figure 4: An example of using data augmentation to address the problem of knowledge conflict.

Secondly, we evaluate the performance of the proposed two-stage mechanism with a large number of conflicting edit requests. In Table 4, we present the Average Probability Difference (Avg-$\Delta_P$) and Target Success Rate (Succ) before and after resolving knowledge conflict. Specifically, we experiment with 1,000 pairs of target knowledge and corresponding conflicting knowledge. A larger Avg-$\Delta_P$ (i.e., output probability of target knowledge minus output probability of conflicting knowledge) and a higher Succ (i.e., the target knowledge is the final output) indicate that the model is more inclined to output the target knowledge, which indicates that knowledge conflict is resolved. As illustrated in Table 4, our two-stage mechanism effectively mitigates the issue.

Table 4: COLLABEDIT utilizes augmented edit requests to mitigate the knowledge conflict.

|  | **Avg-$\Delta_P$** | **Succ** |
|---|---|---|
| Before Resolve | -18.11 | 37% |
| After Resolve | 17.6 | 77.6% |

**Dynamic C can alleviate the knowledge forgetting.** As described in Section 3.3.2, we assume that each client has a set of old edit requests $\mathcal{E}_o$ (initially edited), as well as $m$ sets of new edit requests $\mathcal{E}_n = [\mathcal{E}_{n_1}, \mathcal{E}_{n_2}, \cdots, \mathcal{E}_{n_m}]$. We note that for this experiment, there exists no conflict between $\mathcal{E}_o$ and $\mathcal{E}_n$, which allows us to investigate the effects of knowledge forgetting. As shown in Table 5, we find that *after numerous rounds of editing, the LLMs produce much lower PS and ES for knowledge obtained from $\mathcal{E}_o$ due to the knowledge forgetting.* Under the same condition, we dynamically update the covariance matrix C according to Equation 7 when editing both $\mathcal{E}_o$ and $\mathcal{E}_n$. We observe that the *dynamic C significantly mitigates the issue, with the Score only dropping from 79.03 to 78.15 on GPT-J and MCF.*

Table 5: Dynamic covariance matrix C can alleviate the knowledge forgetting. We gather all the edit requests in each round and apply global KE to edit the global model to study the knowledge forgetting issue. For experiments, we initially use $\mathcal{E}_o$ to edit the global model and sequentially use $m$ sets of aggregated new edit requests, where we set $m$ to a large value (i.e., $m = 1000$). We report the editing performance of old edit requests $\mathcal{E}_o$ before and after $m$ rounds of new editing. GPT-J (6B) and GPT2-XL is used.

| Model | Method | MCF | | | | zsRE | | | |
|---|---|---|---|---|---|---|---|---|---|
| | | NS ↑ | PS ↑ | ES ↑ | Score ↑ | NA ↑ | PA ↑ | EA ↑ | Score ↑ |
| GPT-J | Before $m$ rounds of editing | 57.20 | 96.13 | 99.26 | 79.03 | 28.05 | 88.79 | 92.05 | 51.92 |
| | After $m$ rounds of editing (Immutable C) | 65.14 | 76.94 | 84.58 | 74.68 | 24.21 | 61.05 | 66.22 | 41.21 |
| | After $m$ rounds of editing (Dynamic C) | 58.15 | 91.62 | 97.32 | 78.15 | 26.54 | 79.34 | 84.40 | 48.28 |
| GPT2-XL | Before $m$ rounds of editing | 65.08 | 80.66 | 89.66 | 77.08 | 25.25 | 64.71 | 68.96 | 43.12 |
| | After $m$ rounds of editing (Immutable C) | 64.89 | 60.38 | 69.82 | 64.80 | 25.28 | 50.31 | 53.96 | 38.47 |
| | After $m$ rounds of editing (Dynamic C) | 61.54 | 74.33 | 82.30 | 71.72 | 24.40 | 56.57 | 59.89 | 39.80 |

## 6  THE DISCUSSION ON THE PRIVACY PRESERVING OF COLLABEDIT

This section theoretically and empirically justifies that COLLABEDIT is privacy-preserving via sharing $\mathbf{K}\mathbf{K}^\top$. We begin our justification by defining input keys $\mathbf{K}$ as:

$$\mathbf{K} = [\mathbf{k}_1, \mathbf{k}_2, \cdots, \mathbf{k}_M] \in \mathbb{R}^{d \times M}, \tag{8}$$

where $d$ indicates the dimension of the feature vector and $M$ indicates the number of edit requests.

**Theoretical aspect.** We aim to prove that it is nontrivial to reconstruct the $\mathbf{K}$ given $\mathbf{K}\mathbf{K}^\top$, which is equivalent to proving that given any specific $\mathbf{K}\mathbf{K}^\top$, there exists an infinite number of $\mathbf{K}$ (every $\mathbf{K}$ may involve different $M$) that will lead to the same $\mathbf{K}\mathbf{K}^\top$.

Let's assume there exists a matrix operation $\mathbf{W}' \in \mathbb{R}^{M \times M'}$, which can transform $\mathbf{K}$ into $\mathbf{K}'$ through $\mathbf{K}' = \mathbf{K} \cdot \mathbf{W}'$ and ensure that $\mathbf{K}'\mathbf{K}'^\top = \mathbf{K}\mathbf{K}^\top$. Then we have:

$$\mathbf{K}'\mathbf{K}'^\top = \mathbf{K}\mathbf{W}'^\top(\mathbf{K}\mathbf{W}')^\top = \mathbf{K}(\mathbf{W}'\mathbf{W}'^\top)\mathbf{K}^\top = \mathbf{K}\mathbf{K}^\top, \tag{9}$$

where any orthogonal matrix $\mathbf{W}'$ such that $\mathbf{W}'\mathbf{W}'^\top = \mathbf{I}$ will lead to the $\mathbf{K}'$ which has the same covariance matrix as $\mathbf{K}$. Since there exists (Grove, 2002; Hall & Hall, 2013) an infinite number of the orthogonal matrix $\mathbf{W}'$ that meets the condition of $\mathbf{W}'\mathbf{W}'^\top = \mathbf{I}$ when $M > 1$, we can conclude that it is nontrivial to reconstruct the $\mathbf{K}$ given $\mathbf{K}\mathbf{K}^\top$ from theoretical perspective[3].

**Empirical aspect.** Our objective is to quantify the extent of privacy leakage by recovering the input sequences of edit requests solely based on the observed $\mathbf{K}$ or $\mathbf{K}\mathbf{K}^\top$. Notably, $\mathbf{K}$ compromises the feature embeddings of input sequences, and thus we leverage the SoTA embedding inversion attack, GEIA (Li et al., 2023a), to recover input sequences from their feature embeddings.

For generality, we adopt the same setup as GEIA to recover input sequences. The key idea is to build a powerful attacker model to decode the sequences from embeddings. The privacy leakage is measured by embedding similarity (Cer et al., 2017) between original sequences and recovered sequences in terms of an LLM (e.g., T5-Large (Raffel et al., 2020)). Since we also want to measure the privacy leakage of $\mathbf{K}\mathbf{K}^\top$, we further tailor the attacker model to recover input sequences from $\mathbf{K}\mathbf{K}^\top$. Considering that $\mathbf{K}\mathbf{K}^\top$ is a covariance matrix involving $M$ input sequences, we calculate the maximum embedding similarity between the recovered sequence and any of the $M$ sequences.

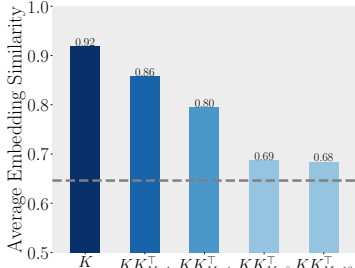

Figure 5: We show the average embedding similarity between recovered sequences (inferred from $\mathbf{K}$ or $\mathbf{K}\mathbf{K}^\top$ involving $M$ sequences) and their ground truths. The grey line is the average embedding similarity between two random text sequences.

Figure 5 shows that sharing $\mathbf{K}$ results in severe privacy leakage as the recovered sequences are close to the original sequences with large embedding similarity. In contrast, with only a small $M$ such as 8, $\mathbf{K}\mathbf{K}^\top$ reduces the embedding similarity to 0.69, which is close to that between two random text sequences (grey line). In other words, the recovered sequence from $\mathbf{K}\mathbf{K}$ is almost irrelevant to any of the $M$ sequences when $M \geq 8$. Therefore, we show that COLLABEDIT achieves privacy-preserving via sharing $\mathbf{K}\mathbf{K}^\top$.

## 7  CONCLUSION AND FUTURE WORKS

In this work, we propose the first collaborative KE framework, COLLABEDIT, which allows multiple parties to jointly edit the knowledge of an LLM without disclosing their private edit requests. In particular, COLLABEDIT leverages the model merging techniques to combine the updates made by each client in their local models. Motivated by the theoretical analysis, we design our framework to be non-destructive, which achieves comparable performance to directly editing a global model using aggregated edit requests. Based on COLLABEDIT, we further provide a remedy toward solving intervention challenges raised in collaborative KE. Interesting future works include: (1) Further improving the performance of KE in collaborative learning scenarios; and (2) Diving deeper into the solutions to fully address intervention challenges in collaborative KE.

---

[3]The clients typically edit multiple requests simultaneously into the LLM and may also apply techniques (e.g., MLE (Li et al., 2024)) to augment their knowledge. Therefore, it is reasonable to assume there are at least 2 edit requests in a single round (or it could be forced in regulation).

ACKNOWLEDGEMENT

This work was partly supported by the NSFC under No. U244120033, 62402418, 62402425, the Key R&D Program of Ningbo under No. 2024Z115, the China Postdoctoral Science Foundation under No. 2024M762829, and the Zhejiang Provincial Priority- Funded Postdoctoral Research Project under No. ZJ2024001, the Research Center for Industries of the Future (RCIF) at Westlake University, and the Westlake Education Foundation.

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

## A   DETAILS OF KNOWLEDGE EDITING IN A SINGLE LLM

**Details of identifying the critical path of MLP layers.**   Following MEMIT (Meng et al., 2023), we apply causal tracing to LLMs (e.g., GPT-2 XL) and identify the critical path of MLP layers to edit. For consistency, we edit the same set of layers $\mathcal{R}$ as MEMIT such as the 13-17th layers of GPT-2 XL.

**Details of the closed form optimization of $\Delta$ for a single layer.**   We optimize the following objective to obtain the optimal weights $\mathbf{W}^*$ of layer $l$:

$$\mathbf{W}^* \triangleq \arg\min_{\hat{\mathbf{W}}} \left( \sum_{i=1}^{n} \left\| \hat{\mathbf{W}}\mathbf{k}_i - \mathbf{m}_i \right\|^2 + \sum_{i=n+1}^{n+|\mathcal{E}|} \left\| \hat{\mathbf{W}}\mathbf{k}_i - \mathbf{m}_i \right\|^2 \right) , \tag{10}$$

where $k_i$ ($1 \leq i \leq n$) indicates the old keys derived from existing knowledge and $k_i$ ($n + 1 \leq i \leq n + |\mathcal{E}|$) indicates the new keys derived from the edit requests $\mathcal{E}$.

Next, we denote $\mathbf{W}$ as the model weights before knowledge editing, $\mathbf{K}_{\text{init}} = [\mathbf{k}_1, \ldots, \mathbf{k}_n]$ as the set of old keys derived from existing knowledge and $\mathbf{K} = [\mathbf{k}_{n+1}, \ldots, \mathbf{k}_{n+|\mathcal{E}|}]$ as the set of new keys derived from the edit requests $\mathcal{E}$. Moreover, $\mathbf{M}_{\text{init}} = [\mathbf{m}_1, \ldots, \mathbf{m}_n] = \mathbf{W}\mathbf{K}_{\text{init}}$ represents the memory values of $\mathbf{K}_{\text{init}}$ that are previously stored and $\mathbf{M} = [\mathbf{m}_{n+1}, \ldots, \mathbf{m}_{n+|\mathcal{E}|}]$ represents the desired memory values of $\mathbf{K}$ that we aim to store. We can solve the Equation (10) by applying *the normal equation*(Strang, 2022):

$$\begin{aligned} \left( \mathbf{W} + \mathbf{\Delta} \right)(\mathbf{K}_{\text{init}}\mathbf{K}_{\text{init}}^\top + \mathbf{K}\mathbf{K}^\top) &= \mathbf{M}_{\text{init}}\mathbf{K}_{\text{init}}^\top + \mathbf{M}\mathbf{K}^\top, \\ \mathbf{W}\mathbf{K}_{\text{init}}\mathbf{K}_{\text{init}}^\top + \mathbf{W}\mathbf{K}\mathbf{K}^\top + \mathbf{\Delta}\mathbf{K}_{\text{init}}\mathbf{K}_{\text{init}}^\top + \mathbf{\Delta}\mathbf{K}\mathbf{K}^\top &= \mathbf{M}_{\text{init}}\mathbf{K}_{\text{init}}^\top + \mathbf{M}\mathbf{K}^\top. \end{aligned} \tag{11}$$

In addition, we define two variables: (1) $\mathbf{C} \triangleq \mathbf{K}_{\text{init}}\mathbf{K}_{\text{init}}^\top$, which represents the covariance matrix of the input keys of existing knowledge. (2) $\mathbf{R} \triangleq \mathbf{M} - \mathbf{W}\mathbf{K}$, which represents the residual error of the new associations when evaluated on the old weights $\mathbf{W}$. Then, we can obtain the closed-form solution of the weight updates $\mathbf{\Delta}$ as:

$$\mathbf{\Delta} = \mathbf{R}\mathbf{K}^\top (\mathbf{C} + \mathbf{K}\mathbf{K}^\top)^{-1}. \tag{12}$$

We compute $\mathbf{C} = \mu \cdot \mathbb{E}_k \left[ \mathbf{k}\mathbf{k}^\top \right]$, where $\mathbb{E}_k \left[ \mathbf{k}\mathbf{k}^\top \right]$ is estimated as an uncentered covariance statistic collected using an empirical sample of vector inputs to the layer (e.g., 100,000 Wikipedia records). $\mu$ is a hyperparameter that balances the weighting of new v.s. old associations (a typical value of $\mu$ is $1.5 \times 10^4$ according to MEMIT).

**Details of the implementation on simultaneously editing multiple layers.**   Previously we focus on illustrating how existing knowledge editing algorithms edit a single layer in the LLM. To simultaneously edit multiple layers of $l \in \mathcal{R}$, existing editing algorithms (e.g., MEMIT (Meng et al., 2023)) firstly obtain the desired output vector $\mathbf{z}_i$ of final layer in $\mathcal{R}$ that can maximize $\Pr \left[ o_i | \mathbf{x} \oplus p(s_i, r_i) \right]$. Then, they spread the whole residual over all the layers in $\mathcal{R}$ by computing partial residual $\mathbf{r}_i^l = \frac{\mathbf{z}_i - \mathbf{W}_i^l \mathbf{k}_i^l}{L - l + 1}$ of each layer, i.e., $l \in \mathcal{R}$. Then, the desired memory value of layer $l$ can be computed as $m_i^l = \mathbf{W}_i^l \mathbf{k}_i^l + r_i^l$ and we can use Equation (12) to edit each layer. For details of the implementation, please also refer to Meng et al. (2023). In this work, we strictly follow their implementation to simultaneously edit multiple layers.

## B   THEORETICAL ANALYSIS OF THE METHODS

For ease of understanding, we will describe knowledge editing for a specific layer $l$ and omit $l$ for brevity. We denote $\mathbf{\Delta}_G$ and $\mathbf{\Delta}_i$ as the weight updates derived from GLOBAL-EDIT and client $i$'s edit. $\mathbf{K}_G$ and $\mathbf{K}_i$ represent the new keys derived from all the edit requests and client $i$'s edits requests. According to Section 3.1, $\mathbf{R}_G$ and $\mathbf{R}_i$ represent the residual errors in the output space of layer $l$ derived from all the edit requests and client $i$'s edits requests, respectively. $\mathbf{C}$ represents the aggregated statistic over the previously stored keys of existing knowledge. We consider the collaborative editing scenario with $N$ clients and each client model has $M$ edit requests.

### B.1 ANALYSIS OF THE NON-DESTRUCTIVE COLLABORATIVE KNOWLEDGE EDITING

Note that $\boldsymbol{\Delta}_i$ and $\boldsymbol{\Delta}_G$ can be computed via (2) as:

$$\begin{aligned}
\boldsymbol{\Delta}_G &= \mathbf{R}_G \mathbf{K}_G^\top (\mathbf{C} + \mathbf{K}_G \mathbf{K}_G^\top)^{-1}, \\
\boldsymbol{\Delta}_i &= \mathbf{R}_i \mathbf{K}_i^\top (\mathbf{C} + \mathbf{K}_i \mathbf{K}_i^\top)^{-1}.
\end{aligned} \tag{13}$$

Following the definitions of $\mathbf{K}$ and $\mathbf{R}$ in Section 3.1, we have:

$$\begin{aligned}
\mathbf{K}_i &= [\mathbf{k}_{i \times (M-1)+1}, \mathbf{k}_{i \times (M-1)+2}, \cdots, \mathbf{k}_{i \times M}], \\
\mathbf{R}_i &= [\mathbf{r}_{i \times (M-1)+1}, \mathbf{r}_{i \times (M-1)+2}, \cdots, \mathbf{r}_{i \times M}], \\
\mathbf{K}_G &= [\mathbf{k}_1, \mathbf{k}_2, \cdots, \mathbf{k}_{N \times M}] = [\mathbf{K}_1, \mathbf{K}_2, \cdots, \mathbf{K}_N], \\
\mathbf{R}_G &= [\mathbf{r}_1, \mathbf{r}_2, \cdots, \mathbf{r}_{N \times M}] = [\mathbf{R}_1, \mathbf{R}_2, \cdots, \mathbf{R}_N].
\end{aligned} \tag{14}$$

Then we have:

$$\mathbf{R}_G \mathbf{K}_G^\top = \mathbf{R}_1 \mathbf{K}_1^\top + \mathbf{R}_2 \mathbf{K}_2^\top + \cdots + \mathbf{R}_N \mathbf{K}_N^\top. \tag{15}$$

According to Equations (13) and (15), we can obtain:

$$\begin{aligned}
\boldsymbol{\Delta}_G (\mathbf{C} + \textstyle\sum_{j=1}^N \mathbf{K}_j \mathbf{K}_j^\top) &= \boldsymbol{\Delta}_G (\mathbf{C} + \mathbf{K}_1 \mathbf{K}_1^\top \cdots + \mathbf{K}_N \mathbf{K}_N^\top) \\
&= \boldsymbol{\Delta}_G (\mathbf{C} + \mathbf{K}_G \mathbf{K}_G^\top) \\
&= \mathbf{R}_G \mathbf{K}_G^\top \\
&= \mathbf{R}_1 \mathbf{K}_1^\top + \mathbf{R}_2 \mathbf{K}_2^\top + \cdots + \mathbf{R}_N \mathbf{K}_N^\top \\
&= \boldsymbol{\Delta}_1 (\mathbf{C} + \mathbf{K}_1 \mathbf{K}_1^\top) + \cdots + \boldsymbol{\Delta}_N (\mathbf{C} + \mathbf{K}_N \mathbf{K}_N^\top) \\
&= \textstyle\sum_{i=1}^N \boldsymbol{\Delta}_i (\mathbf{C} + \mathbf{K}_i \mathbf{K}_i^\top).
\end{aligned} \tag{16}$$

According to the Equation (16), we can finally reach the following conclusion:

$$\boldsymbol{\Delta}_G = \textstyle\sum_{i=1}^N \boldsymbol{\Delta}_i (\mathbf{C} + \mathbf{K}_i \mathbf{K}_i^\top)(\mathbf{C} + \textstyle\sum_{j=1}^N \mathbf{K}_j \mathbf{K}_j^\top)^{-1}. \tag{17}$$

### B.2 ANALYSIS OF THE GAP BETWEEN TWO EDITING METHODS

According to the Equation (17), we obtain the relationship between $\boldsymbol{\Delta}_G$ with $\boldsymbol{\Delta}_i$ as:

$$\boldsymbol{\Delta}_G = \boldsymbol{\Delta}_1 (\mathbf{C} + \mathbf{K}_1 \mathbf{K}_1^\top)\mathbf{A}^{-1} + \cdots + \boldsymbol{\Delta}_N (\mathbf{C} + \mathbf{K}_N \mathbf{K}_N^\top)\mathbf{A}^{-1}, \tag{18}$$

where $A = (\mathbf{C} + \sum_{j=1}^N \mathbf{K}_i \mathbf{K}_i^\top)$. Furthermore, we denote the weight updates derived from the destructive collaborative knowledge editing method using "Task-Arithmetic (TA)" as $\boldsymbol{\Delta}_G'$. We have:

$$\boldsymbol{\Delta}_G' = \lambda \times (\boldsymbol{\Delta}_1 + \boldsymbol{\Delta}_2 + \cdots + \boldsymbol{\Delta}_N). \tag{19}$$

Then, the gap between $\boldsymbol{\Delta}_G$ and $\boldsymbol{\Delta}_G'$ can be calculated as:

$$\begin{aligned}
\boldsymbol{\Delta}_G - \boldsymbol{\Delta}_G' &= \sum_{i=1}^N (\boldsymbol{\Delta}_i (\mathbf{C} + \mathbf{K}_i \mathbf{K}_i^\top)\mathbf{A}^{-1}) - \sum_{i=1}^N \lambda \times \boldsymbol{\Delta}_i \\
&= \sum_{i=1}^N \boldsymbol{\Delta}_i \left[ (\mathbf{C} + \mathbf{K}_i \mathbf{K}_i^\top)(\mathbf{C} + \sum_{j=1}^N \mathbf{K}_i \mathbf{K}_i^\top)^{-1} - \lambda \mathbf{I} \right].
\end{aligned} \tag{20}$$

## C EVALUATION METRICS

### C.1 METRICS FOR MULTI-COUNTERFACT

Multi-CounterFact (MCF) contains an assortment of prompts and texts for evaluating model rewrites. For $(s_i, r_i)$, knowledge editing aims to rewrite the old object $o_i^c$ with the new desired object $o_i$. We use the same metrics as previous works (Meng et al., 2023) for evaluation:

- **Efficacy Success** (ES) is the proportion of cases where the new object $o_i$ exceeds the old object $o_i^c$ in probability:

$$\mathbb{E}_i \left[ \Pr_{\mathcal{M}_\theta} [o_i | p(s_i, r_i)] \geq \Pr_{\mathcal{M}_\theta} [o_i^c | p(s_i, r_i)] \right]. \tag{21}$$

- **Paraphrase Success** (PS) is the proportion of cases where the new object $o_i$ exceeds the old object $o_i^c$ in probability on rephrasings of the original statement:

$$\mathbb{E}_i \left[ \mathbb{E}_{p \in \text{paraphrases}(s_i, r_i)} \left[ \text{Pr}_{\mathcal{M}_{\boldsymbol{\theta}}} \left[ o_i | p \right] > \text{Pr}_{\mathcal{M}_{\boldsymbol{\theta}}} \left[ o_i^c | p \right] \right] \right]. \tag{22}$$

- **Neighborhood Success** (NS) is the proportion of neighborhood prompts (all such prompts have the same old object $o_i^c$) where the model still assigns higher probability to the old object:

$$\mathbb{E}_i \left[ \mathbb{E}_{p \in \text{neighborhood prompts}(s_i, r_i)} \left[ \text{Pr}_{\mathcal{M}_{\boldsymbol{\theta}}} \left[ o_i | p \right] < \text{Pr}_{\mathcal{M}_{\boldsymbol{\theta}}} \left[ o_i^c | p \right] \right] \right]. \tag{23}$$

### C.2 METRICS FOR ZSRE

For the sake of consistency, we report the same three accuracy-based metrics as the previous work (Meng et al., 2023) to evaluate the editing performance on zsRE when using MEMIT (Meng et al., 2023):

- **Efficacy Accuracy** (EA) is the proportion of edits that the model $\mathcal{M}_{\boldsymbol{\theta}}$ recalls with top-1 accuracy. Specifically, an edited model $\mathcal{M}_{\boldsymbol{\theta}}$ should correctly recall the target object $o_i$ with the largest probability given a templated prompt $p(s_i, r_i)$ containing $s_i$ and $r_i$:

$$\mathbb{E}_i \left[ o_i = \arg \max_{o_i'} \text{Pr}_{\mathcal{M}_{\boldsymbol{\theta}}} \left[ o_i' | p(s_i, r_i) \right] \right]. \tag{24}$$

- **Paraphrase Accuracy** (PA) is the accuracy of rephrasings of the original statement:

$$\mathbb{E}_i \left[ \mathbb{E}_{p \in \text{paraphrases}(s_i, r_i)} \left[ o_i = \arg \max_{o_i'} \text{Pr}_{\mathcal{M}_{\boldsymbol{\theta}}} \left[ o_i' | p \right] \right] \right]. \tag{25}$$

- **Neighborhood Accuracy** (NA) is the proportion of neighborhood prompts that the model gets correct for the old object $o_i^c$:

$$\mathbb{E}_i \left[ \mathbb{E}_{p \in \text{neighborhood prompts}(s_i, r_i)} \left[ o_i^c = \arg \max_{o_i'} \text{Pr}_{\mathcal{M}_{\boldsymbol{\theta}}} \left[ o_i' | p \right] \right] \right]. \tag{26}$$

## D ALGORITHM OF OUR COLLABEDIT

---

**Algorithm 1** COLLABEDIT: Non-destructive Collaborative Knowledge Editing

---

**Require:** The number of clients $N$, edit requests $\mathcal{E}_i$ of each client ($1 \leq i \leq N$) where $\mathcal{E}_i = \{(s_{ij}, r_{ij}, o_{ij} | j)\}$, language model $\mathcal{M}_{\boldsymbol{\theta}}$ with weights $\mathbf{W}^l$ of layer $l$, a set of MLP layers to edit $\mathcal{R}$, covariance matrix $\mathbf{C}$ of existing knowledge (optional for direct editing methods, e.g., MEMIT), Hyper-network $\mathcal{H}$ with learnable parameter $\kappa_l$ for layer $l$ (optional for hypernetwork-based editing methods, e.g., MALMEN), a set of prompt templates $\mathcal{P}$.

**Ensure:** Edited language model $\mathcal{M}_{\boldsymbol{\theta}}$ with updated weights $\mathbf{W}^* = \mathbf{W} + \boldsymbol{\Delta}$ of layer $l$.

1: $\boldsymbol{\Delta}_{list} = [\,]$, $\mathbf{KKT}_{list} = [\,]$
2: **for** $i \in \mathcal{N}$ **do**
3: $\quad \boldsymbol{\Delta}_{list}^i, \mathbf{KKT}_{list}^i \leftarrow \text{GetDeltaAndKKT}(\mathcal{E}_i, \mathcal{M}_{\boldsymbol{\theta}}, \mathbf{C}, \mathcal{H}, \mathcal{P})$
4: $\quad \boldsymbol{\Delta}_{list}.append(\boldsymbol{\Delta}_{list}^i)$, $\mathbf{KKT}_{list}.append(\mathbf{KKT}_{list}^i)$
5: **for** $l \in \mathcal{R}$ **do**
6: $\quad \mathbf{A} \leftarrow \mathbf{C}$
7: $\quad \mathbf{A} \leftarrow \kappa_l \mathbf{I}$
8: $\quad$ **for** $i \in \mathcal{N}$ **do**
9: $\quad\quad \mathbf{K}_i^l \mathbf{K}_i^{l\top} = \mathbf{KKT}_{list}[i][l]$, $\boldsymbol{\Delta}_i^l = \boldsymbol{\Delta}_{list}[i][l]$
10: $\quad\quad \mathbf{A} \leftarrow \mathbf{A} + \mathbf{K}_i^l \mathbf{K}_i^{l\top}$
11: $\quad\quad \boldsymbol{\Delta}_i^l \leftarrow \boldsymbol{\Delta}_i^l \times (\mathbf{C} + \mathbf{K}_i^l \mathbf{K}_i^{l\top})$
12: $\quad\quad \boldsymbol{\Delta}_i^l \leftarrow \boldsymbol{\Delta}_i^l \times (\kappa_l \mathbf{I} + \mathbf{K}_i^l \mathbf{K}_i^{l\top})$
13: $\quad\quad \mathbf{W}^{*l} \leftarrow \mathbf{W}^l + \sum_{i=1}^{\mathcal{N}} \boldsymbol{\Delta}_i^l \times \mathbf{A}^{-1}$

---

---

**Algorithm 2** GetDeltaAndKKT

---

1: **procedure** GETDELTAANDKKT($\mathcal{E}_i, \mathcal{M}_{\boldsymbol{\theta}}, \mathcal{H}, \mathbf{C}, \mathcal{P}$)
2:     **for** $s_j, r_j, o_j \in \mathcal{E}_i$ **do**
3:        $L_j \leftarrow \frac{1}{|\mathcal{P}|} \sum_{k=1}^{|\mathcal{P}|} -\log \mathrm{Pr}_{\mathcal{M}_{\boldsymbol{\theta}}} \left[ o_j | \mathcal{P}_k(s_j, r_j) \right]$
4:        **optimize** $\mathbf{z}_j \leftarrow \arg\min_{\mathbf{z}_j} L_j$    ▷ the desired output of modified layers to output $o_j$ given $(s_j, r_j)$
5:        Cache $L_j$
6:     $\boldsymbol{\Delta}_{list} = [], \mathbf{KKT}_{list} = []$
7:     **for** $l \in \mathcal{R}$ **do**
8:        $\mathbf{h}_i^l \leftarrow \mathbf{h}_i^{l-1} + \mathbf{a}_i^l + \mathbf{m}_i^l$
9:        **for** $s_j, r_j, o_j \in \mathcal{E}_{i,j}$ **do**
10:          $\mathbf{k}_i^l \leftarrow \mathbf{k}_i^l = \frac{1}{\mathcal{P}} \sum_{k=1}^{|\mathcal{P}|} \mathcal{P}_k(s_j, r_j)$
11:          $\mathbf{r}_i^l \leftarrow \frac{\mathbf{z}_j - \mathbf{W}^l \mathbf{k}^l}{\mathcal{R}[-1] - l + 1}$
12:          $\mathbf{r}_i^l \leftarrow \mathcal{H}(\mathbf{k}_i^l, \nabla_{\mathbf{k}_i^l} L_j) \mathbf{k}_i^l$
13:        $\mathbf{K}^l \leftarrow [\mathbf{k}_1^l, ..., \mathbf{k}_i^l]$
14:        $\mathbf{R}^l \leftarrow [\mathbf{r}_1^l, ..., \mathbf{r}_i^l]$
15:        $\boldsymbol{\Delta}^l \leftarrow \mathbf{R}^l \mathbf{K}^{l\top} (\mathbf{C}^l + \mathbf{K}^l \mathbf{K}^{l\top})^{-1}$
16:        $\boldsymbol{\Delta}^l \leftarrow \mathbf{R}^l \mathbf{K}^{l\top} (\lambda_l \mathbf{I} + \mathbf{K}^l \mathbf{K}^{l\top})^{-1}$
17:        $\boldsymbol{\Delta}_{list}.\mathrm{append}(\boldsymbol{\Delta}^l)$ , $\mathbf{KKT}_{list}.\mathrm{append}(\mathbf{K}^l \mathbf{K}^{l\top})$
18:     **return** $\boldsymbol{\Delta}_{list}, \mathbf{KKT}_{list}$

---

Table 6: Knowledge conflict can compromise the editing performance of collaborative KE. We denoted $\mathcal{E}$ and $\mathcal{E}'$ in section 5.3. Edit $\mathcal{E}$ indicates that only requests in $\mathcal{E}$ are edited, while Edit $\mathcal{E}$ and $\mathcal{E}'$ indicates that requests in both sets are edited. We evaluate the editing performance of edit requests in $\mathcal{E}$.

| Model | Method | NS ↑ | PS ↑ | ES ↑ | NA ↑ | PA ↑ | EA ↑ |
|-------|--------|------|------|------|------|------|------|
| GPT-J | Edit $\mathcal{E}$ | 57.09 | 96.31 | 99.2 | 5.32 | 69.24 | 91.96 |
| | Edit $\mathcal{E}$ and $\mathcal{E}'$ | 60.59 | 85.43 | 91.18 | 5.33 | 27.83 | 48,84 |
| GPT2-XL | Edit $\mathcal{E}$ | 64.85 | 81.06 | 89.56 | 8.5 | 38.89 | 58.28 |
| | Edit $\mathcal{E}$ and $\mathcal{E}'$ | 63.77 | 69.89 | 78.16 | 7.54 | 15.96 | 24.28 |

Table 7: A summary of scenarios of knowledge conflict.

| Situation | Analysis |
|-----------|----------|
| $m_1 = m_2$ | Two conflicted editing events $e_1$ and $e_2$ are made by the same client. In this case, the client could directly apply knowledge augmentation techniques (e.g., Multi-label Editing Li et al. (2024)) to overwrite its previous knowledge. |
| $m_1 \neq m_2$ and $t_1 = t_2$ | Two conflicted editing events $e_1$ and $e_2$ are made by different clients at the same round of editing. In this case, we need to further develop a two-stage mechanism to solve conflict as illustrated in Section 4.2.2. |
| $m_1 \neq m_2$ and $t_1 \neq t_2$ | Two conflicted editing events $e_1$ and $e_2$ are made by different clients at different rounds of editing. In this case, we need to further develop a two-stage mechanism to solve conflict as illustrated in Section 4.2.2. |

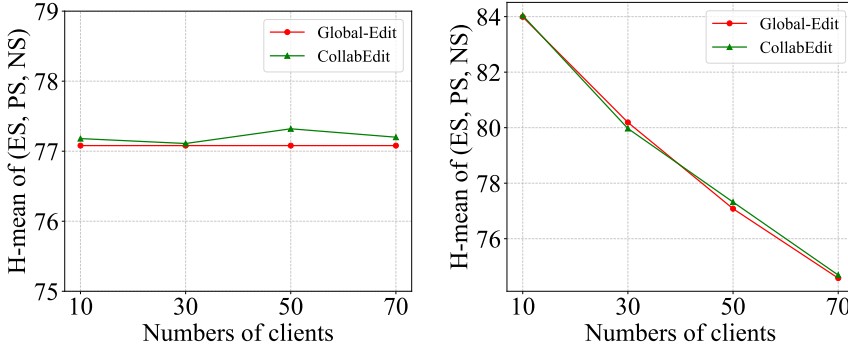

(a) A total number of 5000 edit requests     (b) Every client edits 100 requests

Figure 6: Comparison of global KE (GLOBAL-EDIT) and collaborative KE with different client numbers.

# E CONCRETE EXAMPLE OF GENERATED CONFLICT OBJECT

**Instruction:**

**Prompts for generating a conflict object**

```
{"You're tasked with creating a new benchmark for
↪  conflicts in knowledge clipping. Given a set of
↪  data structure as shown in the example, your
↪  objective is to generate a conflict object.
↪  Specifically:

1) Modify the 'target_new' field to a similar but
↪  different && incorrect answer.
2) Adapt the 'attribute_prompts' accordingly to
↪  reflect the modified 'target_new'.
3) Provide the output in JSON format, containing the
↪  modified 'target_new' and the adjusted
↪  'attribute_prompts'.

Input: {}

Output:"}
```

**Outputs: A conflict object response for case-0 in MCF by GPT-3.5-turbo**

**case-0-conflict:**

```
{
"case_id": 0,
"pararel_idx": 2796,
"requested_rewrite": {
"prompt": "The mother tongue of {} is",
"relation_id": "P103",
"target_new": {
"str": "Spanish",
"id": -1
},
"target_true": {
"str": "French",
"id": "Q150"
},
"subject": "Danielle Darrieux"
},
"paraphrase_prompts": [
"Shayna does this and Yossel goes still and dies.
↪  Danielle Darrieux, a native",
"An album was recorded for Capitol Nashville but
↪  never released. Danielle Darrieux spoke the
↪  language"
],
"neighborhood_prompts": [
"The mother tongue of Léon Blum is",
"The native language of Montesquieu is",
"François Bayrou, a native",
"The native language of Raymond Barre is",
"Michel Rocard is a native speaker of",
"Jacques Chaban-Delmas is a native speaker of",
```

```
            "The native language of François Bayrou is",
            "Maurice Genevoix, speaker of",
            "The mother tongue of François Bayrou is",
            "Melchior de Vogüé, speaker of"
            ],
            "attribute_prompts": [
            "J.\\xa0R.\\xa0R. Tolkien is a native speaker of",
            "The mother tongue of Douglas Adams is",
            "The mother tongue of Paul McCartney is",
            "Elvis Presley is a native speaker of",
            "Barack Obama, speaker of",
            "Douglas Adams, speaker of",
            "Meryl Streep, a native",
            "George Orwell spoke the language",
            "George Washington, a native",
            "Michael Jackson, a native"
            ],
            "generation_prompts": [
            "Danielle Darrieux's mother tongue is",
            "Where Danielle Darrieux is from, people speak the
            ↪  language of",
            "Danielle Darrieux's mother tongue is",
            "Danielle Darrieux was born in",
            "Danielle Darrieux's mother tongue is",
            "Danielle Darrieux's mother tongue is",
            "Danielle Darrieux was born in",
            "Where Danielle Darrieux is from, people speak the
            ↪  language of",
            "Danielle Darrieux was born in",
            "Danielle Darrieux was born in"
            ]
            }
```

**Outputs: Original data of case-0 in MCF**

**case-0-conflict:**

```
            {
            "case_id": 0,
            "pararel_idx": 2796,
            "requested_rewrite": {
            "prompt": "The mother tongue of {} is",
            "relation_id": "P103",
            "target_new": {
            "str": "English",
            "id": "Q1860"
            },
            "target_true": {
            "str": "French",
            "id": "Q150"
            },
            "subject": "Danielle Darrieux"
            },
            "paraphrase_prompts": [
            "Shayna does this and Yossel goes still and dies.
            ↪  Danielle Darrieux, a native",
```

```
                "An album was recorded for Capitol Nashville but
                ↪   never released. Danielle Darrieux spoke the
                ↪   language"
            ],
            "neighborhood_prompts": [
            "The mother tongue of L\u00e9on Blum is",
            "The native language of Montesquieu is",
            "Fran\u00e7ois Bayrou, a native",
            "The native language of Raymond Barre is",
            "Michel Rocard is a native speaker of",
            "Jacques Chaban-Delmas is a native speaker of",
            "The native language of Fran\u00e7ois Bayrou is",
            "Maurice Genevoix, speaker of",
            "The mother tongue of Fran\u00e7ois Bayrou is",
            "Melchior de Vog\u00fc\u00e9, speaker of"
            ],
            "attribute_prompts": [
            "J.\u00a0R.\u00a0R. Tolkien is a native speaker of",
            "The mother tongue of Douglas Adams is",
            "The mother tongue of Paul McCartney is",
            "Elvis Presley is a native speaker of",
            "Barack Obama, speaker of",
            "Douglas Adams, speaker of",
            "Meryl Streep, a native",
            "George Orwell spoke the language",
            "George Washington, a native",
            "Michael Jackson, a native"
            ],
            "generation_prompts": [
            "Danielle Darrieux's mother tongue is",
            "Where Danielle Darrieux is from, people speak the
            ↪   language of",
            "Danielle Darrieux's mother tongue is",
            "Danielle Darrieux was born in",
            "Danielle Darrieux's mother tongue is",
            "Danielle Darrieux's mother tongue is",
            "Danielle Darrieux was born in",
            "Where Danielle Darrieux is from, people speak the
            ↪   language of",
            "Danielle Darrieux was born in",
            "Danielle Darrieux was born in"
            ]
            }
```

