# OpenReview forum: "CollabEdit: Towards Non-destructive Collaborative Knowledge Editing"
_ICLR.cc/2025/Conference — ICLR 2025 Poster_

### Official Review · Reviewer_h2zh · 2024-10-28

**Soundness:** 3
**Presentation:** 3
**Contribution:** 3
**Rating:** 6
**Confidence:** 4

**Summary:**

This paper introduces COLLABEDIT, a framework designed for collaborative knowledge editing (KE) in LLMs within federated learning scenarios. COLLABEDIT allows multiple parties to collaboratively edit the knowledge in LLMs while preserving data privacy, a novel scenario within knowledge editing and federated learning. It addresses three main challenges—knowledge overlap, knowledge conflict, and knowledge forgetting—by implementing a non-destructive model merging technique that aims to achieve performance close to direct global model editing without degrading results. Extensive experiments on GPT-J and GPT2-XL demonstrate the effectiveness of COLLABEDIT, showing improvements over existing approaches in federated scenarios.

**Strengths:**

* The paper identifies and addresses a novel problem of knowledge editing in federated learning for LLMs, a new setting within model editing research.
* The authors propose a straightforward yet effective method—COLLABEDIT—that enables privacy-preserving collaborative editing, which is an essential consideration in multi-party learning scenarios.
* Experiments on GPT-J and GPT2-XL show that COLLABEDIT can substantially improve performance over methods like MEMIT in federated settings, highlighting its practical effectiveness in this new problem space.

**Weaknesses:**

* The need for collaborative knowledge editing within federated LLM may be limited, as large-scale federated LLM scenarios are currently uncommon. This reduces the perceived applicability and impact of the problem being solved.
* The experiments are conducted on older models like GPT-J and GPT2-XL. More recent models such as LLaMA-2, LLaMA-3, or Gemma would provide stronger validation of the proposed method’s efficacy.
* The paper’s structure could benefit from refinement, as some figures and tables (e.g., Figure 3 and Table 4) are misaligned, affecting readability and presentation quality.

**Questions:**

* Why was the setup of editing 10 models with 500 requests (Table 1 and 2) per model not applied consistently in Table3?
* Could you clarify why the MCF dataset was not included in experiments in Table 3? This dataset would likely provide a valuable benchmark for evaluating the framework’s robustness in handling knowledge conflicts.
* In the knowledge overlap experiments, the focus was on the  R value’s  $\ell_2$-norm rather than directly showing the editing method’s performance. How does COLLABEDIT perform when subjected to repeated editing requests for the same knowledge items?

---

> ### Author Response · Authors · 2024-11-23
> **Rebuttal for reviewer h2zh (Part 1)**
>
> Dear reviewer `h2zh`:
>
> Thank you for your review. We would like to address your concerns in detail below.
>
> > **W1:** Large-scale federated LLM scenarios are currently uncommon
> >
>
> **A:** The reviewer might underestimate the research value and accumulated achievements of federated learning [1,2] and collaborative learning [9]. Below are some real-world application examples:
>
> - **Industrial federated** **scenarios: *Tencent's Tianyan Lab*** and ***WeBank*** jointly developed a medical federated learning framework [3]. ***NVIDIA*** introduced ***NVIDIA Clara*** [5] Federated Learning and NVIDIA FLARE SDK [10] for healthcare data privacy. On September 6, 2019, ***WeBank*** and ***Extreme Vision*** launched the first visual federated learning system [6] for industry upgrades.
> - **The application of Federated learning for LLM:**
>     - On 16 Oct. 2023, ***FATE*** [11] facilitates federated learning for large language models.
>     - On 5 Apr. 2024, ***Google*** [12] applied federated LLM in `GBoard` with notable real-word results.
>     - On 11, Oct. 2024, ***Prime Intellect*** [13] announced the first decentralized training of a 10B Parameter Model.
>     - On 5 Nov. 2024, ***Photon*** [7] proposed worldwide federated pre-training of LLMs for data privacy.
>     - Collaborative/Federated learning [7,8,11,12,13] is a highly **promising approach** to address the issues of information silos and the contradiction between local limited computing resources and the unbounded scale of models. Collaborative KE, as compared to local training in FL, offers a more compute- & memory- efficient form of local knowledge editing and holds significant promise.
>
> Finally, we want to emphasize that regardless of whether the real-world applications or scenarios are “common”, the ultimate goal of scientific research is to **solve existing problems** and **cultivate fertile ground for theoretical applications**. Scientific research should precede industrial applications and lay the theoretical foundation for the industry, which may contribute to the development of the industry and lead to the emergence of more applications.
>
> [1] Openfedllm: Training large language models on decentralized private data via federated learning
>
> [2] Federated unlearning: Guarantee the right of clients to forget
>
> [3] Privacy-Preserving Technology to Help Millions of People: Federated Prediction Model for Stroke Prevention
>
> [4] [EGX Platform for Accelerated Computing | NVIDIA](https://www.nvidia.com/en-us/data-center/products/egx/)
>
> [5] [NVIDIA Clara](http://nvidia.com/clara)
>
> [6] FedVision: An Online Visual Object Detection Platform Powered by Federated Learning
>
> [7] Photon: Federated LLM Pre-Training
>
> [8] The Future of Large Language Model Pre-training is Federated
>
> [9] Collaborative learning via prediction consensus
>
> [10] [NVIDIA FLARE](https://nvidia.github.io/NVFlare/)
>
> [11] FATE-LLM: A Industrial Grade Federated Learning Framework for Large Language Models
>
> [12] Prompt Public Large Language Models to Synthesize Data for Private On-device Applications
>
> [13] [INTELLECT–1: Launching the First Decentralized Training of a 10B Parameter Model](https://www.primeintellect.ai/blog/intellect-1)
>
> ---
>
> > **W2:** Additional experiments on recent models such as LLaMA-2, LLaMA-3, or Gemma.
> >
>
> **A: R-Table 2** presents additional experiments of collaborative KE on LLama-3-8B. We use MEMIT as the backend KE algorithm and adopt the default setting in our paper (i.e., 10 clients and 5000 edit requests in total). The experiments show that our `CollabEdit` still achieves non-destructive editing performance on the LLama-3.
>
> **R-Table 2:** Overall editing performance on LLama-3, based on MEMIT. The “Score” serves as the overall metric. (All metrics are better when higher)
>
> | **Method** | NS⬆ | PS⬆ | ES⬆ | **Score**⬆ |
> | --- | --- | --- | --- | --- |
> | **Global-Edit** | 86.62 | 76.07 | 95.66 | 85.36 |
> | **Ties-Merging** | 89.65 | 16.44 | 16.36 | 22.53 |
> | **Task-Arithmetic** | 49.33 | 51.12 | 50.48 | 50.29 |
> | **Simple-Average** | 89.92 | 10.94 | 10.04 | 14.84 |
> | **CollabEdit** | **85.8** | **77.2** | **95.3** | **85.46** |
>
> ---
>
> > **W3:** some figures and tables (e.g., Figure 3 and Table 4) are misaligned
> >
>
> **A:** As mentioned in the **Global Response**, we have adjusted the structure of the paper and highlighted these changes **in orange.**

---

> ### Author Response · Authors · 2024-11-23
> **Rebuttal for reviewer h2zh (Part 2)**
>
> > **Q1:** Why was the setup of editing 10 models with 500 requests (Table 1 and 2) per model not applied consistently in Table3 ?
> **Q2:** Could you clarify why the MCF dataset was not included in experiments in Table 3?
> >
>
> **A:** Our experiments on MALMEN are primarily based on modifying their source code to support our collaborative knowledge editing. Therefore, to ensure the effectiveness of MALMEN, we use a consistent experimental setup as MALMEN.
>
> Table 3 differs from Table 1 and Table 2 in our paper mainly from the following two perspectives:
>
> - **We evaluate MALMEN with 8 models and 125 edit requests per model due to the inherent issue of MALMEN’s codes.**
>     - **Simple experiments with default parameters:** Table 3 was proposed to verify whether our CollabEdit's collaborative editing performance is universally applicable across different KE methods. Therefore, we temporarily conducted a simple `1000=8x125` experiment based on the MALMEN’s codes, which is a relatively default experimental setup for MALMEN’s codes.
>     - **Defects in the source code of the MALMEN:** Due to underlying implementation issues, we are temporarily unable to modify the source code of MALMEN to support collaborative editing with 5000 edit requests. Directly changing the hyperparameters to 5000 edits would result in insufficient memory in the A800 (80G) GPU (it needs retraining the hypernetwork). The optimization of MALMEN's code is not within the scope of our work; we simply aim to validate the effectiveness of our framework on MALMEN. However, we will continue to optimize the MALMEN code in the future to complete the `5000=10x500` experiment and integrate it into our code framework.
> - **We evaluate MALMEN just on the zsRE dataset because its code lacks support for MCF.**
>     - **Additional experiment of MALMEN on the MCF dataset:** The codebase of MALMEN does not support the MCF dataset. According to our re-implementation (i.e., **R-Table 3 and R-Table 4**), MALMEN is not effective on the MCF dataset. Even for global editing, the editing score is lower than 10%. However, we note that `CollabEdit` still achieves non-destructive editing performance as the global editing in this scenario.
>     - **Discussion on the MALMEN’s bad KE performance:** We explain that the poor editing performance of MALMEN on the MCF dataset is due to the need for a large training dataset for the training of hypernetwork. MALMEN achieves relatively good performance on the zsRE dataset because it uses `163,196` records of zsRE as the training set. However, the MCF dataset only contains `20,877` records in total. Even if we split the dataset with a ratio of 9:1 (R-Table2 and R-Table3), the results are still not desirable.
>
> To better illustrate the generalizability of our `CollabEdit`, we further test our framework using the latest KE method—`AlphaEdit` [1]. `AlphaEdit` mitigates the issue of disruption and achieves state-of-the-art editing performance. As shown in **R-Table 5**, our method still exhibits nearly non-destructive collaborative KE performance when using `AlphaEdit` as the backend (The LLM is GPT2-XL).
>
> [1] AlphaEdit: Null-Space Constrained Knowledge Editing for Language Models
>
> **R-Table 3:** Overall editing performance on GPT-J (6B) , based on MALMEN. We edit 8 models and each model will be edited by 125 requests of mcf. The “Score” serves as the overall metric.
>
> | **Method** | **ES**⬆ | **GS**⬆ | **LS**⬆ | **Score**⬆ |
> | --- | --- | --- | --- | --- |
> | **Global-Edit** | 2.63 | 41.11 | 20.21 | 6.60 |
> | **Ties-Merging** | 0.09 | 8.39 | 14.16 | 0.26 |
> | **Task-Arithmetic** | 1.26 | 17.18 | 19.53 | 3.32 |
> | **Simple-Average** | 0.78 | 20.31 | 19.43 | 2.16 |
> | **CollabEdit** | **2.05** | **42.18** | **18.94** | **5.31** |
>
> **R-Table 4:** Overall editing performance on GPT2-XL , based on MALMEN. We edit 8 models and each model will be edited by 125 requests of mcf. The “Score” serves as the overall metric.
>
> | **Method** | **ES**⬆ | **GS**⬆ | **LS**⬆ | **Score**⬆ |
> | --- | --- | --- | --- | --- |
> | **Global-Edit** | 4.49 | 38.47 | 17.18 | 9.77 |
> | **Ties-Merging** | 0.09 | 6.73 | 9.86 | 0.26 |
> | **Task-Arithmetic** | 1.46 | 18.75 | 16.89 | 3.76 |
> | **Simple-Average** | 1.75 | 24.41 | 15.43 | 4.42 |
> | **CollabEdit** | **4.49** | **41.11** | **17.18** | **9.82** |
>
> **R-Table 5:** Overall editing performance on GPT2-XL, based on AlphaEdit. We edit 10 models and each model will be edited by 500 requests of mcf. The “Score” serves as the overall metric.
>
> |  | MCF |  |  |  |
> | --- | --- | --- | --- | --- |
> | **Method** | **NS**⬆ | **PS**⬆ | **ES**⬆ | **Score**⬆ |
> | **Global-Edit** | 65.51 | 85.4 | 97.76 | 80.63 |
> | **Ties-Merging** | 75.42 | 31.97 | 31.72 | 39.44 |
> | **Task-Arithmetic** | 51.74 | 58.04 | 65.64 | 57.92 |
> | **Simple-Average** | 77.62 | 34.71 | 44.32 | 46.68 |
> | **CollabEdit** | **63.61** | **84.08** | **96.04** | **78.89** |

---

> ### Author Response · Authors · 2024-11-23
> **Rebuttal for reviewer h2zh (Part 3)**
>
> > **Q3:** How does COLLABEDIT perform when subjected to repeated editing requests for the same knowledge items ?
> >
>
> A: We have theoretically addressed the issue of knowledge overlap between different rounds. However, in our paper, we have yet to evaluate its impact on the overall performance in a single round. Below is our supplementary experiment:
>
> - **Experimental setup and results:** We conduct additional experiments to investigate the impact of knowledge overlap in **R-Table 6.** Specifically, we randomly sample 2000 edit requests from the MCF dataset as the non-overlapped dataset. Then, we randomly sample another k edit requests (e.g., k=10) and repeat them to construct an overlapped dataset with 2000 edit requests.
>     - Motivation: We aim to explore the impact of the overlapped dataset on the editing performance of the non-overlapped dataset in a single round.
>     - Results: **R-Table 6** reports the editing performance of non-overlapped dataset with/without additional repeated records. We can observe that even if we repeat k edit requests multiple times to incorporate a large overlapped dataset (i.e., 2000), the editing performance of the non-overlapped dataset is surprisingly not affected.
> - **Discussion:** However, it is important to note that we did not consider more complex scenarios across multiple rounds, where unexpected effects might occur. Our theoretical derivation in Section 4.2.1 suggests that by evaluating the L2 norm of the residuals $\mathbf{R}$ before editing, we can identify whether there are overlapping parts between the previous editing and the current one. Such a mechanism can avoid the waste of computational resources and the emergence of uncontrollable issues.
>
> **R-Table 6:**  Supplementary experiment for Knowledge Overlap in a single round
>
> | **Method** | NS⬆ | PS⬆ | ES⬆ | **Score**⬆ |
> | --- | --- | --- | --- | --- |
> | w/ repeated records | 68.35 | 98.25 | 99.9 | 86.16 |
> | w/o repeated records | 68.21 | 98.25 | 100 | 86.11 |

---

> ### Author Response · Authors · 2024-11-27
> **A follow-up message about the rebuttal for the CollabEdit paper**
>
> Dear reviewer `h2zh`：
>
> We hope this message finds you well.
>
> We are writing to kindly inquire about **the status of your feedback on our recent rebuttal**. We understand that your time is valuable, and we greatly appreciate the effort you have already put into reviewing our manuscript. Your insights are crucial to the improvement of our work, and **we are eager to address any remaining concerns you may have**.
>
> If there are any additional questions or clarifications needed from our side, please **do not hesitate to let us know**. Since the discussion phase has been extended, we hope to take advantage of this additional valuable time to **engage in more in-depth exchanges with you**.
>
> Thank you once again for your time and consideration. We look forward to hearing from you soon.
>
> Best regards,
>
> Authors of `CollabEdit`

---

### Official Review · Reviewer_hEBc · 2024-10-31

**Soundness:** 2
**Presentation:** 3
**Contribution:** 3
**Rating:** 6
**Confidence:** 3

**Summary:**

This paper addresses the generalization of knowledge editing within the collaborative learning setting, with a focus on ensuring privacy while modifying the knowledge of large language models (LLMs). The authors propose a novel approach by sharing $KK^{T}$, an intermediate weight associated with the keys of edited knowledge, instead of naively sharing and averaging weights, which is theoretically proven to be resistant to attacks. The experiments conducted demonstrate the effectiveness of the proposed approach.

**Strengths:**

+ The paper tackles an important problem of generalizing knowledge editing to collaborative learning settings where privacy is a critical concern.
+ The authors provide a compelling theoretical analysis of the limitations of naive weight sharing and introduce the concept of sharing $KK^{T}$, which is proved to be difficult to attack in the traditional privacy-aware setting.
+ The experiments conducted seem to effectively demonstrate the effectiveness of the proposed method.

**Weaknesses:**

- It is not surprising to see the destructive performance of direct fed-average for knowledge editing, as edits individual client are naturally diluted when models are averaged, although I appreciate the formal mathematical treatment of the issue.
- While knowledge conflict is identified as a key challenge, the paper addresses it in a rather ad hoc manner compared to other challenges, which are supported by theoretical analysis.
- My biggest concern is on the privacy part of the model. Although the authors propose to share $K^{T}K$ and providing theoretical proof of its resistance to attacks, the paper does not fully address the new privacy challenges faced by LLMs. If the edit is successful, the new knowledge can be easily prompted out from the LLMs by simply asking questions. This is especially convenient given that most knowledge editing tasks involve only the editing of factual knowledge. Therefore, the traditional privacy methods may not suffice in the LLM case, and further exploration in preserving privacy for knowledge editing is needed.

**Questions:**

Please refer to my summary of my weaknesses.

---

> ### Author Response · Authors · 2024-11-23
> **Rebuttal for reviewer hEBc (Part 1)**
>
> Dear reviewer `hEBc`:
>
> Thank you for your review. We would like to address your concerns in detail below.
>
> ---
>
> > **W1**: The destructive performance of fed-average for knowledge editing is not surprising
> >
>
> **A:** Thank you for your appreciation of our mathematical contribution! Below are some responses to the mentioned weakness:
>
> - **Trying to solve rather than just reveal the problems:** In this work, we **pioneer the exploration** of collaborative KE and compare the performance of existing collaborative learning methods (e.g., fed-average) for KE. We emphasize that revealing the destructive performance of these methods is not the main focus of this work. Instead, we aim to tackle this performance gap with our designed theoretical framework to promote the practical applications of collaborative KE.
> - **Our contribution:**
>     - **Tackling an unsolved problem:** Traditional collaborative learning methods lead to significant degradation in editing performance, highlighting the difficulty of collaborative KE.
>     - **Laying the Groundwork:** Our work systematically investigates this important yet challenging problem based on our theoretical framework. In particular, we propose `CollabEdit`, the first non-destructive collaborative KE method that allows multiple parties to jointly edit the knowledge of LLMs with guaranteed performance.
>     - **Comprehensive Analysis:**  We identify three key problems tailored to this paradigm and propose approaches to (effectively) mitigate the performance drop (e.g., applying a dynamic covariance matrix to address knowledge forgetting). We believe this work can inspire a broad range of further exploration in this direction.
>
> ---
>
> > **W2:** Knowledge Conflict is addressed in a rather ad hoc manner.
> >
>
> **A:** Knowledge conflict poses a significant challenge in the context of collaborative KE.
>
> - **The primary challenge lies in the subjective and personalized nature of conflict resolution**. When conflicts arise between edits made by different clients, determining which edits to retain is inherently **ambiguous**. The decision should be tailored to specific application scenarios. For example, one might prioritize edits from clients with higher urgency or those who have made more substantial contributions.
>     - In this work, we aim for a **general and objective** approach. Specifically, we propose to utilize strategies such as `FCFS` (First Come, First Served) to resolve conflict and apply data augmentation (e.g., prompt rephrasing) to preserve the desired edit requests. We believe this approach, while straightforward, is effective and adaptable in tackling the issue of knowledge conflict in various scenarios.
> - **Another significant challenge lies in identifying knowledge conflicts in a privacy-preserving manner**. Our framework prioritizes privacy by allowing conflicts to occur initially and then addressing them in subsequent rounds based on specific criteria (e.g. `FCFS`). Moreover, we employ data augmentation techniques to mitigate the effects of the knowledge conflict.
>
> Though there might be other methods to resolve the challenges, they involve mechanism design and are beyond the scope of this manuscript. We hope that our approach can serve as a catalyst for further research and inspire more insightful discussions in this direction.

---

> ### Author Response · Authors · 2024-11-23
> **Rebuttal for reviewer hEBc (Part 2)**
>
> > **W3:** The new knowledge can be easily prompted out from the LLMs by asking questions.
> >
>
> **A:** Thanks for this intriguing question. However, we have noticed that the concerns raised by the reviewer contain some misunderstandings regarding the contributions of our CollabEdit. We will clarify them as follows:
>
> **1. The privacy risk of the reviewer’s concern is different from that of** `CollabEdit`**.**
>
> - **The risk in our** `CollabEdit`**:** The issue mentioned by the reviewer is indeed one that LLMs should address. This issue is independent of the privacy risks that our `CollabEdit` focuses on. In our threat model, we focus on whether we can ensure the privacy of client-edited data in collaborative scenarios and, based on this, also ensure the privacy of the client's identity (we cannot determine who edited the knowledge).
> - **The risk in reviewer’s concern:** The reviewer's concern, on the other hand, is about whether model weights might leak private data. To be honest, all KE methods may have this risk, and to address this, it might be more appropriate to firstly investigate strategies such as access control or safety alignment in just a single model context. These topics are out of the scope of our work, and we leave for future researchers to explore the intersection of these two areas (e.g., whether performing KE on a safety-aligned model would reduce its security).
>
> **2. The reviewer assumes stronger adversary capabilities: attackers may have to guess the edit requests.**
>
> - The reviewer assumes that the attacker knows client-edited data and can craft targeted queries (questions) to test whether some specific data has been updated.
>     - However, it is non-trivial to extract the data due to the lack of access to edit requests (Please refer to the proof regarding the privacy guarantees of $KK^{\top}$ in Section 6 of paper).
>     - Additionally, `CollabEdit` ensures that the attacker cannot identify which user edits a specific piece of knowledge due to the anonymization, which protects the identity of a client.
>     - In conclusion, the attacker can hardly gain any direct or related knowledge about the edits.
> - Since the attacker cannot extract any information from the edit requests ($KK^{\top}$), he/she can only use random queries to the old and new models. **With a large search space of** **edit requests**, it is challenging for a third party to **guess** what has been edited during the KE (In other words, the cost of the attack for a specific edit is extremely high).
>     - This is particularly important when KE is utilized for machine unlearning [1]. Specifically, machine unlearning often involves edit requests with private information (e.g., ID numbers), which, unless previously leaked, would be nearly impossible for an average person to accurately guess.
>
> **Summary:** the attack proposed by the reviewer follows a different threat model and assumes different adversary capabilities.
>
> [1] On Knowledge Editing in Federated Learning: Perspectives, Challenges, and Future Directions

---

> > ### Comment · Reviewer_hEBc · 2024-11-23
> > **Thanks for the response.**
> >
> > I would like to thank the authors for the detailed response to address my concerns. I have adjusted my scores accordingly. All the best.

---

> ### Author Response · Authors · 2024-12-02
> **Thank you for timely response and increasing scores!**
>
> Dear reviewer `hEBc`:
>
> As the discussion phase is coming to an end, we would like to send this additional message to express our gratitude to you.
>
> You provided constructive and thoughtful review with us and acknowledged the efforts we made during the rebuttal phase （we are very pleased that our rebuttal addressed your concerns）.
>
> We are also extremely grateful for your timely feedback and, most importantly, your increasing scores to `(6)`!!! The concerns reviewer `hEBc` raised have also helped us think about deeper issues within and beyond this work. We hope that our discussion and this work can jointly inspire some new insights and research regarding collaborative editing or trustworthy AI.
>
> Once again, we deeply appreciate your thoughtful and timely feedback as well as the increased scores!!!
>
> Best regards,
>
> Authors of `CollabEdit`

---

### Official Review · Reviewer_jVrX · 2024-11-01

**Soundness:** 3
**Presentation:** 3
**Contribution:** 3
**Rating:** 6
**Confidence:** 3

**Summary:**

The paper investigates the collaborative knowledge editing (KE) for large language models (LLMs). It identifies three primary challenges in this domain: knowledge overlap, knowledge conflict, and knowledge forgetting. The authors propose a framework called COLLABEDIT, which utilizes a non-destructive model merging mechanism to aggregate knowledge edits from multiple parties while maintaining performance and privacy.

The framework aims to mimic the optimal global editing behavior without the significant performance drops associated with existing destructive methods. Through extensive experiments on canonical datasets, the authors demonstrate that COLLABEDIT outperforms traditional approaches, addressing the identified challenges effectively.

**Strengths:**

COLLABEDIT allows for non-destructive knowledge editing, which prevents significant performance drops that are common in traditional methods

The framework is versatile and can integrate existing knowledge editing methods, providing a comprehensive solution to collaborative KE challenges

Empirical results show that COLLABEDIT outperforms existing destructive baselines, demonstrating superior editing performance even with a large number of edits

**Weaknesses:**

The non-destructive merging mechanism may introduce additional complexity in implementation compared to simpler, traditional methods.

Its scalability in large collaborative environments or with numerous clients may need further exploration.

More experiments on different LLMs could benefit the demonstration of the effectiveness of the proposed method.

**Questions:**

See above.

---

> ### Author Response · Authors · 2024-11-23
> **Rebuttal for reviewer jVrX**
>
> Dear reviewer `jVrX`:
>
> Thank you for your review. We would like to address your concerns in detail below.
>
> ---
>
> > **W1:** The non-destructive merging mechanism may introduce additional complexity in implementation
> >
>
> **A:** In our experiment, we use three model merging algorithms as baselines: *Task Arithmetic (TA)*, *Simple Average (SA)*, and *TIES-merging*. Below, we provide a detailed analysis of each method’s computational complexity, including our `CollabEdit` (refer to **R-Table 0**).
>
> Suppose that there are $N$ clients, the updates $\Delta$ have a dimension of $[v, k]$ (v and k in a similar scale).
>
> - SA and TA calculate averages across all the models, so their time complexity is relatively small: $O(N \times v \times k)$, and the space complexity is $O(N \times v \times k)$.
> - The TIES-Merging includes “Trim”, “Elect”, and “Merge” phases. The “Trim” phase has a time complexity of $O(N \times v \times k \times \log(N \times v \times k))$. The “Elect” and “Merge” phase has a time complexity of $O(N \times v \times k)$. Therefore, the overall time complexity is $O(N \times v \times k \times \log(N \times v \times k))$. The space complexity is also $O(N \times v \times k)$.
> - For CollabEdit, due to the matrix multiplication and matrix inversion operations, it has a time complexity of $O(N \times k^3)$. The space complexity is $O(N \times k \times k)$ because of $KK^{\top}$.
>
> **R-Table 0:** Overall computational complexity for different merging methods.
>
> | **Method** | Time complexity | Space complexity |
> | --- | --- | --- |
> | **Ties-Merging** | $O(N \times v\times k \times log(N \times v \times k))$ | $O(N \times v \times k)$ |
> | **Task-Arithmetic** | $O(N \times v \times k)$ | $O(N \times v \times k)$ |
> | **Simple-Average** | $O(N \times v \times k)$ | $O(N \times v \times k)$ |
> | **CollabEdit** | $O(N \times k^3)$ | $O(N \times k \times k)$ |
>
> In summary,
>
> - Our method incurs only a minimal increase in time complexity to achieve non-destructive collaborative KE. The time complexity can be significantly reduced by leveraging **GPU acceleration for matrix operations**, such as matrix inversion in $KK^{\top}$. As a result, the actual time overhead will be quite small (e.g. **a few seconds in the scenario with 10 clients**).
> - In addition, as the GPU memory is extremely restricted, the space complexity of `CollabEdit` can be further reduced by merging the updates sequentially, which results in a complexity of $O(2 \times v \times k)$.
>
> ---
>
> > **W2:**  `CollabEdit`’s scalability in large collaborative environments or with numerous clients
> >
>
> **A:** We lack access to large-scale collaborative environments, such as industrial-level collaborations, to comprehensively test our framework. However, we evaluate the sensitivity of our framework to the number of clients within a simulated collaborative system.
>
> Specifically, **R-Table 1** compares the editing performance of Global-Editing with that of `CollabEdit` under various numbers of clients. We assume each client edits 100 edit requests in total.
>
> - The results show that regardless of the number of clients, `CollabEdit` consistently achieves similar editing performance as that of Global Editing. Reviewer can also refer to the new **Figure 6** in the appendix of our paper.
> - This highlights the non-destructive nature of our framework and shows its generalizability across diverse scenarios.
>
> **R-Table 1: Overall KE scores of CollabEdit and Global-Edit in scenarios with different clients**
>
> |  | CollabEdit | Global-Edit |
> | --- | --- | --- |
> | **10 clients** | 84.04 | 83.99 |
> | **30 clients** | 79.97 | 80.19 |
> | **50 clients** | 77.32 | 77.08 |
> | **70 clients** | 74.69 | 74.58 |
>
> ---
>
> > **W3:** More experiments on different LLMs
> >
>
> **A:** R-Table 2 presents additional experiments of collaborative KE on LLama-3-8B. We use MEMIT as the backend KE algorithm and adopt the default setting in our paper (i.e., 10 clients and 5000 edit requests in total). The experiments show that our `CollabEdit` still **achieves non-destructive editing performance on the LLama-3.**
>
> **R-Table 2:** Overall editing performance on LLama-3, based on MEMIT. The “Score” serves as the overall metric. (All metrics are better when higher)
>
> | **Method** | NS⬆ | PS⬆ | ES⬆ | **Score**⬆ |
> | --- | --- | --- | --- | --- |
> | **Global-Edit** | 86.62 | 76.07 | 95.66 | 85.36 |
> | **Ties-Merging** | 89.65 | 16.44 | 16.36 | 22.53 |
> | **Task-Arithmetic** | 49.33 | 51.12 | 50.48 | 50.29 |
> | **Simple-Average** | 89.92 | 10.94 | 10.04 | 14.84 |
> | **CollabEdit** | **85.8** | **77.2** | **95.3** | **85.46** |

---

> ### Author Response · Authors · 2024-11-27
> **A follow-up message about the rebuttal for the CollabEdit paper**
>
> Dear reviewer `jVrX`：
>
> We hope this message finds you well.
>
> We are writing to kindly inquire about **the status of your feedback on our recent rebuttal**. We understand that your time is valuable, and we greatly appreciate the effort you have already put into reviewing our manuscript. Your insights are crucial to the improvement of our work, and **we are eager to address any remaining concerns you may have**.
>
> If there are any additional questions or clarifications needed from our side, please **do not hesitate to let us know**. Since the discussion phase has been extended, we hope to take advantage of this additional valuable time to **engage in more in-depth exchanges with you**.
>
> Thank you once again for your time and consideration. We look forward to hearing from you soon.
>
> Best regards,
>
> Authors of `CollabEdit`

---

### Author Response · Authors · 2024-11-23
**Global Response**

Dear Reviewer `jVrX` , `hEBc` and `h2zh` :

We sincerely thank the reviewers for their insightful feedback. We are delighted that the reviewers acknowledged that ***the tackled problems*** are important and novel (Reviewers `jVrX`,`hEBc`,`h2zh`), the collaborative editing ***performance*** of our `CollabEdit` is superior (Reviewers `jVrX`,`hEBc`,`h2zh`), our `CollabEdit` is ***privacy-ensured*** (Reviewers `hEBc`,`h2zh`), the ***experiment*** is effective and substantial (Reviewers `hEBc`,`h2zh`).

Furthermore, our contributions to proposing ***a versatile framework*** (Reviewer `jVrX`), introducing a ***new problem space with practical effectiveness*** (Reviewer `h2zh`), providing ***compelling theoretical analysis*** (Reviewer `hEBc`), the ***comprehensive solutions*** to collaborative KE challenges (Reviewer `jVrX`) are acknowledged.

### **Summary of Contribution and Novelty**

Our work stands out through the following key contributions and innovations:

1. **A Novel Paradigm with Insightful Findings:**
    - To our knowledge, we ***are the first*** to propose the collaborative KE paradigm (***a new problem*** space including naive collaborative KE baselines, Global-Edit, and our `CollabEdit`), which obtains ***practical effectiveness*** and are recognized by Reviewers `h2zh`.
    - Our study introduces ***important and novel problems*** in this novel paradigm, which are also recognized by Reviewers `jVrX`,`hEBc`,`h2zh`.
2. **A Novel Framework with Superior Collaborative KE performance:**
    - we propose the first non-destructive collaborative KE framework with superior ***Collaborative KE performance*,** which are recognized by Reviewers `jVrX`,`hEBc`,`h2zh`.
    - Our `CollabEdit` is ***versatile***, allowing nondestructive integration of existing KE methods and providing insights into ***the solution of three challenge***, as are recognized by Reviewers `jVrX`.
3. **Compelling Theoretical Analysis for Novel problems**
    - We identify ***the performance gap*** between the naive collaborative KE method and the upper bound performance (i.e., GLOBAL-EDIT) through ***formal mathematical analysis*,** which are recognized by Reviewers `hEBc`.
    - Provide ***comprehensive solutions*** to novel collaborative KE challenges, as recognized by Reviewers `jVrX`.
    - Prove the ***privacy-ensured*** nature of our `CollabEdit` with ***compelling theoretical analysis*,** which are also recognized by Reviewers `hEBc`,`h2zh`.
4. **Thorough and effective Experimental Validation**:
    - Our empirical results demonstrate the ***effectiveness*** of our proposed framework compared with baselines and that of ***the novel solutions*** to three challenges based on our `CollabEdit`, which are recognized by Reviewers `jVrX`,`hEBc`,`h2zh`.
    - Our discussions shed light on ***future research*** for collaborative KE.

### **Other Modifications**

- Following some suggestions from the reviewers, we have adjusted the structure of the paper and added some new experimental figures. We have highlighted these changes **in orange**, so the reviewers can easily locate the updated parts:
    1. **Structure Adjustment:** **Figure 3**, **4**, and **5**.
    2. **New Figure:** **Figure 6** in the appendix.
- We have also provided responses to each reviewer's comments regarding the weaknesses and questions raised. (W indicates weakness and Q indicates question)

We hope these responses can address the reviewers' concerns. If you find them helpful, we would be most grateful if you would **consider raising your scores** in a manner. We would also appreciate it if you could inform us **whether our responses adequately address your concerns**. We are open to further discussion or providing additional explanations as needed. Thank you very much again for your thoughtful review and help in improving the paper. We appreciate your time and consideration.

Best regards,

Authors of `CollabEdit`

---

### Meta-Review · Area_Chair_9X4B · 2024-12-21

**Metareview:**

In this paper, the authors explore collaborative knowledge editing in LLMs and propose a framework that leverages non-destructive model merging to enable knowledge editing from multiple parties while preserving both performance and privacy. Reviewers agreed that the proposed framework is effective, the theoretical analysis is robust, and the experiments are thorough. There were discussions regarding the framework's complexity, scalability, and the need for additional experiments with diverse LLMs. Overall, I believe this paper exceeds the acceptance threshold.

**Additional Comments On Reviewer Discussion:**

The authors addressed reviewers' concerns by providing a complexity analysis and conducting simulated experiments to evaluate scalability with a large number of clients and diverse LLMs. They also added further descriptions to highlight the significance of the work. Overall, the responses are satisfactory.

---

### Decision · Program_Chairs · 2025-01-22

Accept (Poster)